# Influence of nitrogen fertilizer rate and variety on tef *[Eragrostis tef (Zucc.) Trotter]* nutritional composition and sensory quality of a staple bread *(Injera)*

**Hayelom Berhe Dagnaw** [1,2]*, **Ashagrie Zewdu Woldegiorgis**[2], **Kebebew Assefa Kebede**[3]

1 Department of Food Science and Nutrition, Ethiopian Institute of Agricultural Research, Addis Ababa, Ethiopia, 2 Center for Food Science and Nutrition, Faculty of Natural and Computational Science, Addis Ababa University, Addis Ababa, Ethiopia, 3 Ethiopian Institute of Agricultural Research, Plant Breeder, Addis Ababa, Ethiopia

* hayelom.berhe8@gmail.com

**Data Availability Statement:** All relevant data are within the paper.

## Abstract

In Ethiopia, tef is one of the major staple crops used as a basic raw material for food development such as stable bread called injera. Studies reported that imbalanced N fertilizer reduces the overall deliciousness of grains. Nowadays nitrogenous fertilizers are considered as the source of variation for the quality of injera, Ethiopian traditional flat bread. Therefore, a field experiment was conducted to assess the effects of N fertilizer rates (0, 30, 60, 90, and 120 kg N ha$^{-1}$) on grain nutrition and sensory quality of injera of three tef varieties of (*Kora*, *Boset*, *and Asgori*). The experiment was conducted in main cropping season in Randomized Complete Block Design with three replications and Di-ammonium Phosphate was used in the same dose. Crop attribute parameters were determined using standard methods. Sensory quality and color of injera were determined by panelists and injera eye software respectively. Results showed that only protein content increased with nitrogen rates, while carbohydrate decreased significantly at ($P < 0.05$). Kora at the control plot (K0) had better color, flavor, texture, and taste values of injera, but they decreased with nitrogen rates. Injera from white tef varieties had a better acceptance as compared with Asgori red tef variety. Injera eye software indicated that the color of injera was significantly affected by varieties. Kora had a higher (55.74) lightness value followed by Boset (54.71), and Asgori (51.26). Injera from the Asgori variety had a maximum red color. *Kora* and *Boset* had higher yellow color on the control plot, but for Asgori it increased with the nitrogen rate.

## Introduction

Tef *(Eragrostis tef (Zucc) Trotter)* is Ethiopian originated grass species [1–4]. Tef is the most common Ethiopian staple crop which is cultivated the same as other varieties such as wheat. Both maturity period and geographical locations have significant variations on tef cultivars. Tef is grown during the main season and it is very resistant to various environmental

**Funding:** The first author of this research article is Hayelom Berhe Dagnaw. I am working at Ethiopian Institute of Agricultural Research, Addis Ababa, Ethiopia. I graduated my M.Sc. degree in food Science and Nutrition from Addis Ababa University, Ethiopia. On the other hand, the second author (supervisor) of this research article was Ashagrie Zewdu Woldegiorgis (Ph.D.) with the position of associate professor. He works at Addis Ababa University, center for Food Siena and Nutrition. The third author (supervisor) of this research article was Kebebew Asefa (Ph.D.), plant breeder, with the position of professor (senior researcher). He works at Ethiopian Institute of Agricultural Research, Addis Ababa, Ethiopia. The funders of this research article were Ethiopian Institute of Agricultural Research (EIAR) and Addis Ababa University (AAU). The URL of each funder website (EIAR) and AAU are, www.eiar.gov.et, and www.aau.edu.et, respectively. The funders had no role in study design, data collection and analysis, decision to publish, or preparation of the manuscript.

**Competing interests:** The authors have declared that no competing interests exist.

conditions such as drought and waterlogging [5, 6]. It is a cereal crop used as a raw material for product development and it accounts for about 15% of calories [7–9].

Poor soil fertility hinders to tef productivities in Ethiopia. Currently the average national productivity of tef is 1.664 tons per hectare [10] which is very low as compared to other cereal crops [11]. *Mwangi* [11], reported that the use of inorganic fertilizer is critical to increasing yield. Nowadays using nitrogen fertilizer becomes as essential component to improve the yield production of cereal crops such as tef [12]. A study conducted using different levels of nitrogen fertilizer with twelve rice cultivars stated that; grain yield decreased as nitrogen fertilizer levels increased [13]. Excessive vegetative growth and waterlogging significantly affects tef grain yield.

The most important constraint in tef production is its inherent low productivity. This has, among others, been due to the inability of farmers to use the required quantities of mineral nutrients and the use of imbalanced chemical fertilizer applications [14]. However, an application of optimum fertilizer rates for specific soil types greatly contributes to yield enhancement. Organized studies should be conducted under varying conditions and in various regions to determine the fertilizer requirements of tef for optimizing yield. This indicated that applications of optimum fertilizer rates have a better contribution for optimum yield productions which should be source of all indispensable nutrients. However, this may vary with soil type and weather condition of the area. Therefore, plant nutrition depends on the availability and uptake of macro and micronutrients contained in the soil [15].

Nitrogen is a major mineral element used in agricultural practices. Studies reported that application of chemical fertilizer for different crop types may have an effect on their yield production [16]. Organoleptic quality and biological values of plant proteins are decreased as a result of excess nitrogen fertilizer [17, 18]. Gu et al. [13], stated that, as the N fertilizer rates are increased, the overall palatability of rice was decreased. The palatability of the food also affected as the protein content of the cultivar increased with nitrogen fertilizer rates. Grain quality and tastiness were negatively correlated with high protein content. However in contrast to this, Gu et al. [13] reported that the lower concentrations of amylose improve cooking and eating qualities with protein content at high nitrogen rates.

The interaction effects of organic or inorganic chemical fertilizers with soil on grain yield and grain nutrition crops have been studied. However, those studied were not focused on sensory qualities of cereal-based food products such as injera, Ethiopian traditional flat bread. On the other hand, there is some variation in the quality of injera from the last decades. Therefore, this study mainly focused on the influence of nitrogen fertilizer rate and variety on tef nutritional composition and sensory quality of staple bread (injera).

## Materials and methods

### Study area

The study was conducted during the main cropping seasons at Debre Zeit Agricultural Research Center (DZARC) from August-November. DZARC is located at 47 km to the south-east of Addis Ababa. The experimental site is characterized by heavy black soil which is the dominant soil type having high water retention capacity by its nature. The place is located at 8˚ 44'; N latitude and 38˚58'; E longitude at an altitude of 1860 meters above sea level and receives an annual average rainfall of about 832 mm.

### Soil sample characterization

Pre-plant soil samples (0–15 cm) were taken using the two ways diagonal in uniform field. After the soil samples were collected, it was mixed and made ready for recommended

parameter analysis. The total nitrogen content of the soil sample was determined by Kjeldhal method, FOSS kjeltec 8400, which follows digestion and distillation. 1 g of air dried soil sample was taken into digested tube. After that, mixture of catalyst and boiling granules and 15 ml concentrate sulfuric acid were added. Then the mixture was digested and distilled. Finally, total protein content was determined after titration with hydrochloric acid [19]. On the other hand the, available phosphorous (P) was determined using Uv-vis spectrophotometer [20], Cary-60, Malaysia. The total organic matter of the pre-plant soil matter was determined using titration method with potassium dichromate as reducing agent [21]. The soil pH was determined using a potentiometer method, W instrument ltd. For the determination of soil pH, 1:1 ratio of air dried soil sample < 2 mm with distilled water was used. Then 10 mL of de-ionized water were added and mixed well using glass road and allowed to stand for 30 minutes and stirred every 10 minutes. Finally pH value was measured using the electrode. Electrical conductivity was measured using a conductivity meter, W instrument ltd. For the determination of electrical conductivity, after the suspension sit was filtered using vacuum pump with suction filtration. Then, the electrical conductivity was measured using the conductivity cell from the filtrate solution.

## Experimental design

The experimental design was a randomized complete block design (RCBD) in a factorial combination of N fertilizer with three replications. The fertilizer treatments were five N levels (0, 30, 60, 90, and 120 kg ha$^{-1}$) with the same dose of triple super phosphate (TSP) (10 kg ha$^{-1}$), as a source of P which is the recommended one for Vertisol of DZARC. The source of N was urea. A full dose of P and half of N was applied at planting while the remaining N dose was applied 30 days later when the plant was in the active vegetative growth stage. The experimental plot size was 5 m × 5 m with a total area of 47 m × 37 m. The distance between plots and blocks was 1 and 1.5 m, respectively. An improved three tef variety of Kora (DZ-Cr-438RIL133B, white color), Boset (DZ-Cr-409 RIL50d, white color), and Asgori (Dz-01-99, red color) were used. All fertilizers were applied as row-side bands. The seeds amounting to 37.5 g per plot based on the recommended seeding rate of 15 kg ha$^{-1}$ were sown by hand drilling in the rows.

## Grain flour preparation

For the preparation of flour, the grain was cleaned and sieved using 1 mm sieve size and the flour was prepared using a milling machine having the model number of EN5501, cyclone sample mill, USA-AID with 0.5 mm of sieve size.

## Injera preparations

Using tef flour and water as ingredients, injera was prepared according to Boka and Yetneberk [22, 23]. 200 g tef flour were mixed with 180 mL water using starter culture and kneaded and mixed well for 5 min. The starter culture was prepared from each sample for individual dough. The dough was fermented at 48 h for 28–30˚C. After the dough was fermented, 100 mL of water were added and the batter was fermented for 2–3 h at 28–30˚C and waited till foam and bubbles were formed. After the bubbles were formed, 500 mL of the batter were poured in a curricular manner on 50 cm diameter of hot clay griddle *mitad* and back cover for about 2 min. Then after two mins, Injera was removed by lifting it off from the hot griddle.

## Image analysis

The color of the injera for scored values for the required parameters was determined after 24 h the injera were baked. To brighten the sample two parallel fluorescent lamps were used. To

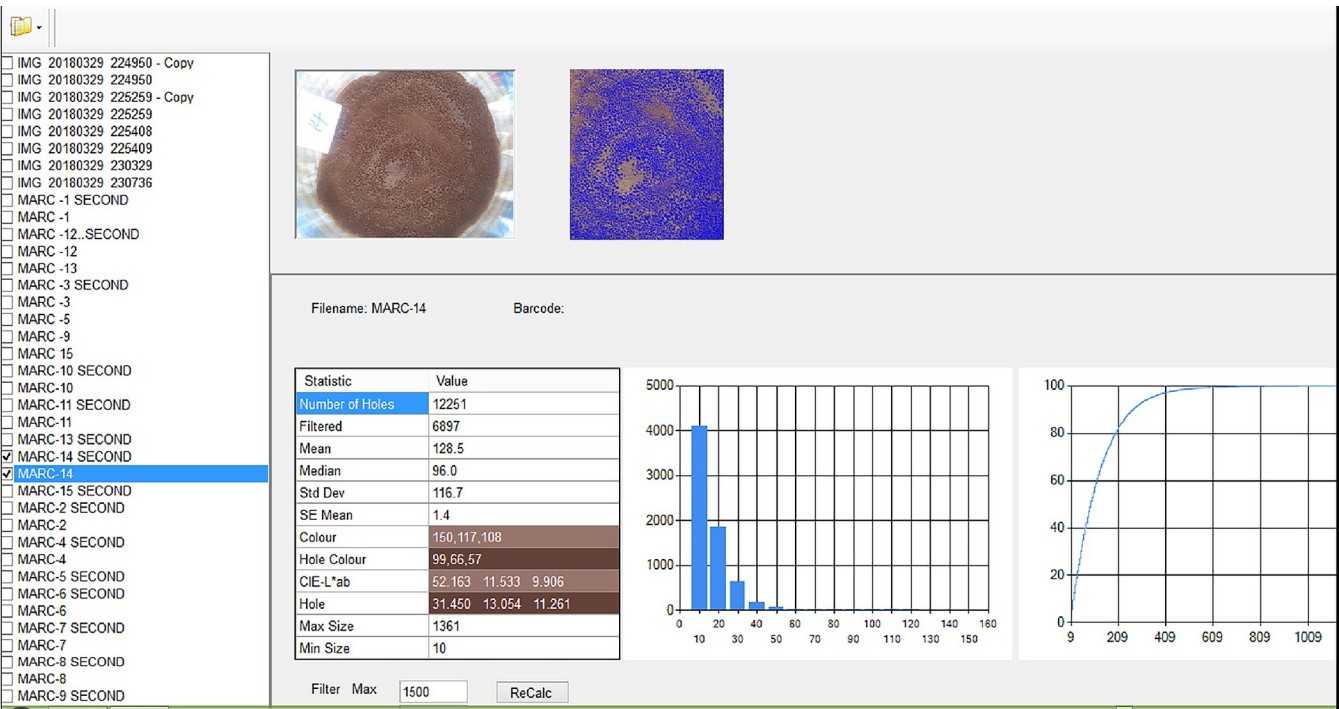

**Fig 1. Image analysis of injera using injera software** [24]. [https://onlinelibrary.wiley.com/doi/abs/10.1002/cche.10252].

obtain a uniform light intensity; the samples were located at 10 cm overhead at the angle 45˚C of the sample plane. The room was switched off in order to protect the light reflection and then the developed injera was removed. The camera was located vertically at a distance of 45 cm from the injera sample and images were captured on the camera setting without flash. For each formula, a total of two injera were captured. Finally, the images were transferred to a computer hard disk and opened with injera eye software (Fig 1). The color (CIE L *a*b) of injera were measured using digital image of the sample visualized and software (Fig 1) [24] which connected with computer. Two surface points from each injera sample were taken by cutting using a cutter having closely related maximum size and standard deviations of injera samples. Therefore, the number of holes (holes of injera and filtered eyes) and Color of injera (lightness (L*, scale 0 to 100), a* (redness), and b* (blue-yellow) color) were determined using injera eye software.

## Laboratory analysis

### Proximate analysis

Proximate composition was determined according to the official method [25]. Moisture content was determined according to [25] official method. Crucibles were cleaned and dried using oven dry method at 105˚C for 1 h and weighed (W1). Using dried crucibles, 2 g dried sample were weighed (W2) and dried at 135˚C for 1 h and re-weighed after cooled (W3). Finally, moisture content of sample flour was calculated using difference of fresh sample and dried samples with crucibles, respectively.

The ash content of tef flour was determined using the official method [25]. Porcelain dishes were placed in a muffle furnace for 30 min at 550˚C. The dishes were cooled in desiccators (with granular silica gel) for some mins at room temperature and weighed **(W1).** 3 g of tef

flour were weighed **(W2).** The weighed samples were charred using hot plate with increased temperature under fume-hood. Then dishes with sample (charred) were placed inside the muffle furnace at 550°C for 5 h. After the time finished the crucible were cooled in desiccator for 1h and reweighed **(W3).** Ash contents of samples were determined using difference of fresh sample and dried samples with crucibles, respectively.

The crude protein content of tef flour was determined through Kjeldhal method according to official method of AOAC, number 979.09 [25]. 0.5 g of dried samples was added in to digestion tubes. Then 15 mL of concentrated sulfuric acid and two tablets of 1000 kjeltabs Cu/3, 5 mixture of catalyst were added and digested at 420°C for 1 h. The distillation and titration process were carried out using a solution having 40% sodium hydroxide as receiver alkali solution and 0.1N of hydrochloric acid as titrant solutions and deionized water. During this analysis ammonium iron (II) sulphate hexa hydrate (0.15 g) with theoretical nitrogen values of ammonium sulphate (21.09%) was used to check the recovery. Therefore, the percentage of protein content was calculated by the digital machine using the conversion factors of 6.25 in triplicate form.

The crude fat contents of tef flour were determined according to the official method [25]. **3** g of the samples were weighed using thimble and covered with purified cotton. 50 mL of n-hexane was used as extraction solvent. The sample with the solvent was placed in the soxtec extractor for 1 h. After 1 h the extra or residual solvent was evaporated using oven dry method at 103°C and the pure extracted fat was cooled in desiccators and weighed. Then a percentage of crude fat content was determined based on the extracted and the fresh sample.

The amount of carbohydrate content of sample was determined by difference, which was done by subtracting the sum percentage of moisture, ash, crude protein, and crude fat contents, from 100 percent. The energy content of tef grain was obtained based on the values of crude protein, crude fat, and total carbohydrate which were multiplied with factors of 4, 9, and 4, respectively and reported in kilocalories per 100 g.

## Determination of mineral content

Mineral contents of Iron (Fe) and calcium (Ca) tef grain were determined using Atomic absorption spectrophotometer method [26] of Agilent technologies, 200 series AA. 5 g flour samples were ashed using maffle furnace at 550°C for five hours. The ashed samples were moistened using distilled water and dissolved in 5 mL HCl and filtered in to 100 mL as final volume. Finally the solution was marked with distilled water and the metal concentrations were determined after the standard solutions were prepared.

Standard solution of Fe and Ca: First 100 ppm of individual metal standard concentrations was prepared from their respective stock solutions (1000 mg/L). Then five series of standard working metal solution 4, 8, 12, 16, 20 for Fe and 1.2,24, 3.6, 4.8, 6.0 for Ca were prepared in 50 ml using deionized water. Using Agilent technology (200 series AA) the concentrations of standard and sample solution were determined using the stand formula.

On the other hand, the totals P from tef flour sample were determined based on the dry-ashing procedure by measuring the absorbance of phosphomolybdate blue method AACC [26]. 3 g of tef flour was measured. A sample portion was added to 30–50 mL crucibles. Crucibles were placed in a cool muffle furnace with increase in temperature gradually to 550°C and ashing continued for 4 to16 h. The cooled samples were moistened using distilled water and dissolved in 5 mL of 12 N hydrochloric acid (HCl) and added drop wise of 12 N HCl until effervescence becomes completed. Then the solution was evaporated to dryness, occasionally stirring with a glass rod. Then 15 mL of 6M hydrochloric acid were added to the residue followed about 120 mL of distilled water.

The solution was stirred using the glass rod and covered the beaker with a watch-glass. Then the solution was gently boiled and maintained at boiling point until no more ash can be seen to dissolve. The mixture solution was filtered using ash-free filter paper and collected in to 250 mL volumetric flask. The beaker was washed and filtered by 5 mL of hot 6N HCl and filled with deionized water. From the filtrate solution 5 mL of the aliquot of the dissolved ash was taken into a 100 mL volumetric flask and 10 mL ammonium vanadomolybdate reagent was added and diluted with deionized water. 50 ppm P stock solutions were prepared by weighing 0.219 g of potassium dihydrogen phosphate ($KH_2PO_4$) in 1L volume with deionized water. Standard solutions containing 0.5, 1.0, 1.5, 2.0 and 2.5 ppm P were prepared from the stock solution including the blank solution. Finally the absorbance of sample and blank were recorded after 30 minutes at 460 nm using UV-Vis Spectrophotometer) Cary-60, Malaysia. By plotting calibration curve using standards, the concentration of P in the sample was determined.

## Determination of phytic acid content

Phytic acid content was determined using modified method [27]. About 0.1g of dried flour sample was extracted with 100 mL 2.4% HCl for 1 h at an ambient temperature and centrifuged (3000 rpm/30 min). The supernatant solution was used for phytate estimation. About 2 mL of Wade reagent were added to 3 mL of the sample solution and centrifuged. Then the absorbance of phytic acid was determined at 500 nm using UV-Vis spectrophotometer.

Working standard solutions were prepared from 300 ppm phytic acid (analytical grade sodium phytate salt) using 0.2N HCl. 90 ppm was used as intermediate solution and 5, 9, 18, 27 and 36 ppm were prepared from the intermediate solution. Then 3 and 2 mL of working solution and Wade reagent were added in to 15 mL of centrifuge tubes and mixed using vortex mixer. Distilled water was used as a blank solution. The absorbance for standard, blank solutions and then the sample solution were measured at 500 nm. The final concentration of phytic acid was determined using the given Eq below (1):

$$\text{Phytic acid content in } \mu g/g \text{ in} = \frac{[\text{CAb} - \text{As}) - \text{intercp}]}{\text{Slope} \times \text{W} \times 3} \times 10 \qquad (1)$$

## Determination of tannin content

Condensed tannin content was analyzed by vanillin-HCl method of Price et al. [28]. The Vanillin-HCl reagent was prepared by mixing equal volume of 8% concentrated HCl in methanol and 1% Vanillin in methanol. The solution of the reagent was mixed just before use. 1.0 g sample was placed in small conical flask. Then 10 mL of 1% concentrated HCl in methanol were added in to 15 mL plastic tubes. Samples were shaked for 24 h using mechanical shaker (IKA[R] AS 130.1, USA) at room temperature. Then the sample was centrifuged at 1000 rpm for 5 min. About 1 mL of the supernatant solution was pipetted into another test tube and mixed with 5 mL of vanillin—HCl reagent and waited for 20 min.

Series of working solutions of 0.15, 0.2, 0.4, 0.6, 0.8 and 1 mL of stock solution (40 mg of D-catechin in 100 mL of methanol) were taken and adjusted to 1 mL with 1% HCl in methanol. Then 5 mL of vanillin-HCl reagent were added and after 20 min the reaction was completed. Finally, the absorbance was measured using Uv-vis, Cary-60, Malaysia at 500 nm.

Then tannin content was expressed as catechin equivalent using the Eq below (2).

$$\text{Tannin (mg/g)} = \frac{[(As - Ab) - Intercept] \times 10}{Slope \times d \times W} \qquad (2)$$

Finally, the total and filtered number of eyes and color of injera were determined using injera eyes software [24].

## Statistical analysis

All the parameters were measured in triplicates and the results are reported as mean ± standard error (S.E). Injera eye software was used for image analysis of injera. And the Statistical analyses were done using SPSS 20.0 window evaluation version program with Duncan's multiple post-hoc test with two-way ANOVA. The differences between pairs of means were evaluated on the basis of 95% confidence interval at ($p \leq 0.05$) level of significance difference.

## Results and discussion

### Soil physicochemical properties of the experimental site

The physiochemical properties of composite soil sample before planting are presented in (Table 1). The result indicated that soil pH affects the nutrient availability and toxicity. The current study is in line with [29], which is reported from Forth and Ellis, (1997). The soil pH (6.5) showed that the experimental soil type is classified as slightly acidic soil. The available P also below the threshold levels of the tropical soils [30, 31]. A study [32], the total percentage of nitrogen is 0.124% and the response of tef to nitrogen fertilizer proved that there was limited available nitrogen in the experimental soil and it is likely be stored in organic matter and clay minerals. The total mean percentage of organic carbon (1.41%) content of the experimental site was comparable to soils in the semiarid regions. Due to the lower number of organic materials applied to the soil and complete removal of the biomass from the field, most cultivated soils in Ethiopia are poor in organic matter [33].

## Tef grain as influenced by nitrogen fertilizer rates and varieties

### Moisture content

The moisture content of tef grain recorded from the various fertilizer treatments was in the ranges reported by Bultosa G. (2007) as, 9–11%. Hence, the moisture contents for brown seeded Asgori variety (DZ-01-99) and the white seeded tef varieties (DZ-01-196) ranged from 12.14 to 13.00% and 11.99–12.23%, respectively [34]. On the other hand, for N rate (over all means of tef varieties), the moisture content at 60 kg ha$^{-1}$ (1013%) had significant variation as compared to 90 (9.97%) and 120 kg ha$^{-1}$ N rate (9.973%) (Table 2).

### Ash content of tef grain

The ash content of tef grain ranged from 1.99% to 3.16%. Bultosa [35], reported that the maximum ash content of Asgori tef variety was 3.16%, whereas the ash content of the present study is 2.42% for the same variety. This suggested that the applications of nitrogen fertilizer could not alter the contents of inorganic elements.

**Table 1. Soil physico-chemical properties of the experimental site and its pre-planting soil sample (0–15 cm depth) analysis.**

| S.S (0-15cm) depth | EC (ds/cm) 1:1 | P$^{H}$ (H$_2$O) 1:1 | %OC | %OM | %TN | Ava. P (ppm) |
|---|---|---|---|---|---|---|
| Block -1 | 0.181 | 6.83 | 1.51 | 2.61 | 0.130 | 7.212 |
| Block -2 | 0.226 | 6.37 | 1.39 | 2.40 | 0.120 | 7.205 |
| Block-3 | 0.215 | 6.31 | 1.41 | 2.43 | 0.122 | 6.928 |
| Mean ± SEM | 0.207±0.170 | 6.50 ± 0.093 | 1.44 ± 0.037 | 2.48 ± 0.066 | 0.124 ± 0.003 | 7.12 ± 0.094 |

EC = electrical conductivity, $_p$H = power of hydrogen ion, OC- = organic carbon, Om = organic matter, TN = total nitrogen, Ava. P = Available phosphorus, S. S = Soil sample and SEM = standard errors of means.

**Table 2. Nutritional composition of tef grain as influenced by tef varieties and N fertilizer rates.**

| Treatments | Parameters | | | | | | |
|---|---|---|---|---|---|---|---|
| VWR | %Moisture | %Ash | %Protein | %Crude fat | % fiber | %Carbo hydrate | Energy (Kcal) |
| K0 | 9.88[a] ± 0.033 | 2.48[a] ± 0.025 | 9.97[a] ± 0.587 | 2.08[a] ± 0.050 | 2.86[a] ± 0.025 | 75.59[a] ± 0.659 | 361.00[a] ± 0.343 |
| K30 | 9.83[a] ± 0.088 | 2.50[a] ± 0.191 | 9.50[a] ± 0.469 | 2.16[a] ± 0.028 | 3.46[a] ± 0.032 | 76.00[a] ±0.512 | 361.47[a] ± 0.68 |
| K60 | 10.13[a] ± 0.058 | 2.38[a] ± 0.134 | 10.61[a] ± 0.242 | 2.13[a] ± 0.062 | 2.60[a] ± 0.045 | 74.75[a] ± 0.424 | 360.60[a] ± 0.337 |
| K90 | 9.97[a] ± 0.033 | 2.41[a] ± 0.017 | 10.39[a] ± 0819 | 2.24[a] ± 0.026 | 3.07[a] ± 0.104 | 74.99[a] ± 0.791 | 361.65[a] ± 0.471 |
| K120 | 9.73[a] ± 0.122 | 2.53[a] ± 0.184 | 11.70[a] ± 0.094 | 2.20[a] ± 0.029 | 3.03[a] ± 0.110 | 73.77[a] ± 0.416 | 361.98[a] ± 0.0503 |
| **G.MK** | **9.909 ± 0.107** | **2.46 ± 0.112** | **10.434 ± 0.654** | **2.162 ± 0.048** | **3.004 ± 0.145** | **75.02 ± 0.689** | **361.34 ± 0.537** |
| B0 | 9.87[a] ± 0.067 | 2.58[a] ± 0.210 | 9.57[a] ± 0.029 | 2.55[a] ± 0.362 | 3.40[a] ± 0.054 | 75.42[a] ± 0.605 | 362.94[a] ± 0.746 |
| B30 | 10.28[a] ± 0.033 | 2.29[a] ± 0.038 | 10.35[a] ± 0.319 | 2.15[a] ± 0.050 | 2.50[a] ± 0.025 | 74.93[a] ± 0.371 | 360.46[a] ± 0.274 |
| B60 | 9.91[a] ± 0.200 | 2.32[a] ± 0.092 | 10.57[a] ± 0.600 | 2.09[a] ± 0.013 | 2.48[a] ± 0.012 | 75.10[a] ± 0.698 | 361.56[a] ± 0.801 |
| B90 | 9.70[a] ± 0.058 | 2.40[a] ± 0.159 | 11.51[a] ± 0.492 | 2.14[a] ± 0.009 | 2.74[a] ± 0.011 | 74.25[a] ± 0.701 | 362.27[a] ± 0.830 |
| B120 | 9.81[a] ± 0.029 | 2.23[a] ± 0.092 | 11.21[a] ± 0.100 | 2.10[a] ±0.048 | 2.70[a] ± 0.102 | 74.59[a] ± 0.245 | 362.66[a] ± 0.279 |
| **G.MB** | **9.91 ± 0.177** | **2.37 ± 0.129** | **10.40 ± 0.506** | **2.16 ± 0.167** | **2.76 ± 0.103** | **74.83 ± 0.518** | **362.08 ± 0.740** |
| A0 | 9.99[a] ±0.088 | 2.30[a] ± 0.015 | 8.99[a] ± 0.159 | 2.24[a] ± 0.015 | 3.73[a] ± 0.114 | 76.47[a] ± 0.416 | 362.00[a] ± 0.920 |
| A30 | 10.00[a] ± 0.088 | 2.39[a] ± 0.029 | 10.58[a] ± 0.537 | 2.21[a] ± 0.053 | 2.92[a] ± 0.101 | 74.82[a] ± 0.654 | 361.46[a] ± 0.429 |
| A60 | 10.07[a] ± 0.219 | 2.45[a] ± 0.152 | 10.79[a] ± 0.182 | 2.17[a] ± 0.015 | 3.35[a] ± 0.113 | 74.51[a] ± 0.157 | 360.71[a] ± 0.466 |
| A90 | 9.60[a] ± 0.088 | 2.41[a] ± 0.113 | 10.98[a] ± 0.065 | 2.16[a] ± 0.026 | 2.84[a] ± 0.054 | 74.85[a] ± 0.193 | 362.74[a] ± 0.649 |
| 1230 | 9.86[a] ± 0.120 | 2.52[a] ± 0.120 | 10.42[a] ± 0.229 | 2.18[a] ± 0.037 | 2.72[a] ± 0.049 | 75.02[a] ± 0.519 | 361.39[a] ± 0.545 |
| **G.MA** | **9.904 ± 0.151** | **2.42 ± 0.096** | **10.67 ±0.485** | **2.19 ± 0.033** | **3.11 ± 0.115** | **75.14 ± 0.568** | **361.66 ± 0.849** |
| **Means of varieties (over all N rates)** | | | | | | | |
| Kora | 9.91[a] ±0.056 | 2.46[a] ± 0.053 | 10.45[a] ± 0.183 | 2.16[a] ± 0.045 | 3.00[a] ± 0.149 | 75.02[a] ± 0.227 | 361.34[a] ± 0.328 |
| Boset | 9.92[a] ±0.056 | 2.36[a] ± 0.053 | 10.64[a] ± 0.183 | 2.22[a] ± 0.045 | 2.76[a] ± 0.149 | 74.86[a] ± 0.227 | 361.98[a] ± 0.328 |
| Asgori | 9.91[a] ± 0.056 | 2.42[a] ± 0.053 | 10.35[a] ± 0.183 | 2.19[a] ± 0.045 | 3.11[a] ± 0.149 | 75.13[a] ± 0.227 | 361.66[a] ± 0.328 |
| **G.MV** | **9.91 ± 0.076** | **2.41 ± 0.108** | **10.48 ± 0.529** | **2.19 ± 0.107** | **2.96 ± 0.307** | **75.00 ± 0.219** | **361.66 ± 0.807** |
| **N rates (Over all means of tef varieties)** | | | | | | | |
| 0 | 9.98[ab] ± 0.138 | 2.46[a] ± 0.127[a] | 9.51[c] ± 0391 | 2.29[a] ± 0.219 | 2.41[a] ± 0.012 | 75.83[a] ± 0.568 | 361.98[a] ± 0.783 |
| 30 | 10.02[ab] ± 0.115 | 2.40[a] ± 0.112[a] | 10.14[bc] ± 0448 | 2.17[a] ± 0.041 | 2.22[a] ± 0.001 | 75.25[ab] ±0.560 | 361.13[a] ± 0.514 |
| 60 | 10.19[ab] ± 0.186 | 2.38[a] ± 0.118[a] | 10.66[ab] ± 0.341 | 2.13[a] ± 0.38 | 2.37[a] ± 0.011 | 74.79[b] ± 0.442 | 360.96[a] ± 0.892 |
| 90 | 9.79[b] ± 0.084 | 2.41[a] ± 0.098[a] | 10.96[a] ± 0.555 | 2.18[a] ± 0.03[4] | 2.53[a] ± 0.018 | 74.70[b] ± 0.572 | 362.22[a] ± 0.638 |
| 120 | 9.89[b] ± 0.110b | 2.43[a] ± 0.129[a] | 11.14[a] ± 0.398 | 2.18[a] ± 0.036 | 2.38a ± 0032[a] | 74.46[b] ± 0.423 | 362.01[a] ± 0.607 |
| **G.MR** | **9.97 ± 0.107** | **2.41 ± 0.113** | **10.48 ± 0.544** | **2.19 ± 0.104** | **2.61 ± 0.51** | **75.00 ± 0.568** | **0.731** |

Values are the means of triplicate experiments. Results are reported as mean ± S.EM, VWR = Varieties of tef with N Rates (kg N ha$^{-1}$), GMK, GMB, G.MA and G. MR = grand means of Kora, Boset and Asgori tef varieties, and N rates, respectively. Kora (K0, K30, K60, K90, K120), Boset (B0, B30, B60, B90, B120), and Asgori (A0, A30, A60, A90, 120) combining with 0, 30, 60, 90 and 120 N rates (kg N ha$^{-1}$) for the treatment number of 1 to 15., and S. EM = Standards of errors of means. Means in the same column and within the same treatment category followed by different letter are significantly different as judged by LSD at p < 0.05. **Note: G.M of Kora, Boset and Asgori varieties are considered as means of varieties (over all N rates).**

## Protein content of tef grain

The analysis of variance indicated that the protein content was highly significantly (P < 0.01) affected by the average mean of N fertilizer rates, but not by means of varieties (Table 2). Average means of nitrogen rates indicated that protein content ranged between 9.51% -11.14%, but not much deviated from that reported by Bultosa [35], (8.7% - 11%). However, nitrogen is the basic protein component, thus increasing of protein contents were un-expected one. This indicated that nitrogen fertilizer rates have significant effects on the crude protein contents of tef varieties.

## Crude fat content

The analysis of variance indicated that; crude fat content of the grain was not affected by tef varieties and N fertilizer rates (Table 2). The range of crude fat content of tef grain was found between 2.16 to 2.22% and this was in line with Bultosa [35] (2.0 to 3.0%). The fat accumulations of tef varieties were uniform. The endosperm and embryo were growth by giving optimal conditions to physiological process as the same as corn affected by organic fertilizers [36].

## Crude fiber

The range of crude fiber content was found between 2.76 to 3.11%, which is in line with the report of Baye [37].

## Total carbohydrate and energy content

Means of N rates had highly significant ($P < 0.05$) effects on carbohydrate (CHO) content (Table 2). Nitrogen fertilizer levels (over all means of tef varieties) and 0 kg ha$^{-1}$ (the control plot that is without N fertilizer) has maximum CHO content as compared to the remaining treatments. CHO was decreased as nitrogen fertilizer level increased and it is inversely related with the protein content.

# Effects of nitrogen fertilizer rates and varieties on the anti-nutritional factor of tef

## Phytic acid content

The phytic acid content was highly significantly ($P < 0.001$) affected by the interaction effects of both nitrogen rates and tef varieties. Phytic acid content from control plots was comparable with those receiving different rates of N fertilizer (Table 3). This is because phytate is a primary P storage in the plant, and; a higher phytic acid is expected with fertilization. According to Baye [37], the phytate content of tef varieties were found between 682 and 1374 mg 100 g$^{-1}$ on a dry weight basis. The current result of phytic acid content (98.2–114.3 mg 100 g$^{-1}$) is not in line with Baye [37]. This difference may be due to the reason of methodology difference or the complex form of P with the experimental soil which leads to reduce the phytic acid availabilities.

## Tannin content of tef grain

The tannin content of tef flour was highly significantly ($P < 0.001$) affected only by varieties (Table 3). The average mean of tef varieties indicated that, both white seeds of Kora and Boset had maximum (29.04) and (20.19) mg 100 g$^{-1}$ tannin contents, respectively. On the other hand, brown seeded Asgori variety had 16.89 mg CE 100 g$^{-1}$ tannin contents and this is almost similar to reports of Baye [37] and *Gebreab* [38]; and this may be probably due to genetic differences of varieties. Nitrogen fertilizer levels and average means significantly affected the tannin content of tef varieties (Table 3).

# Effects of nitrogen fertilizer rates on mineral contents of tef grain

## Iron (Fe)

The mineral content of Iron (Fe) was significantly ($P < 0.05$) affected by varieties and the interaction with N fertilizer rates (Table 4). The range of Fe contents for both white seeded (Kora and Boset) tef varieties were found between 25.85 and 34.58 mg 100 g$^{-1}$, respectively; but for Asgori between 23.33 to 37.39 mg 100 g$^{-1}$. The present results are in line with those of Baye

**Table 3. Means of phytic acid and tannin content of tef grain as influenced by different N fertilizer rates and varieties.**

| Treatments | Parameters (mg 100 g$^{-1}$) | |
|---|---|---|
| VWR | Phytic acid | Tannin |
| G0 | 103.7[b] ± 1.49 | 31.46[a] ± 1.021 |
| K30 | 114.3[ab] ± 6.659 | 29.25[a] ± 0.865 |
| K60 | 101.1[b] ± 0.495 | 28.49[a] ± 0.803 |
| K90 | 98.2[b] ± 0.192 | 27.88[a] ± 0.154 |
| K120 | 108.8[ab] ± 3.668 | 28.12[a] ± 0.828 |
| **G.MK** | **105.22 ± 2.255** | **29.04 ± 1.08** |
| B0 | 104.3[b] ± 2.045 | 16.86[a] ±0.109 |
| B30 | 98.4[b] ± 0.785 | 20.17[a] ± 0.901 |
| B60 | 123.3[a] ± 10.991 | 24.22[a] ± 0.683 |
| B90 | 108.0[ab] ± 3.078 | 21.63[a] ± 0.913 |
| B120 | 109.1[ab] ± 4.009 | 18.05[a] ± 0.798 |
| **G.MB** | **108.62 ± 4.458** | **20.17 ± 1.002** |
| A0 | 114.0[ab] ± 7.961 | 22.10[a] ± 0.867 |
| A30 | 100.7[b] ± 0.959 | 13.87[a] ± 0.147 |
| A60 | 114.6[ab] ± 7.00 | 16.29[a] ± 0.128 |
| A90 | 103.1[b] ± 4.049 | 15.83[a] ± 0.106 |
| A120 | 108.6[ab] ± 4.146 | 16.34[a] ± 0.108 |
| **G.MA** | **108.20 ± 4.245** | **16.87[a] ± 0.149** |
| **Means of varieties (over all N rates)** | | |
| Kora | 105.22[a] ± 1.05 | 29.04[a] ± 1.044 |
| Boset | 108.61[a] ± 1.505 | 20.19[b] ± 1.044 |
| Asgori | 108.19[a] ± 1.505 | 16.89[c] ± 1.044 |
| **G.MV** | **107.34 ± 1.501** | **22. 04 ± 0.904** |
| **N rates (Over all means of tef varieties)** | | |
| 0 | 109.10[a] ± 0.588 | 15.50[a] ± 0.865 |
| 30 | 99.75[a] ± 0.459 | 13.73[a] ± 0.943 |
| 60 | 112.73[a] ± 6.412 | 14.73[a] ± 1.007 |
| 90 | 108.88[a] ± 5.001 | 12.90[a] ± 0.982 |
| 120 | 106.27[a] ± 4.102 | 14.41[a] ± 0.932 |
| **G.MR** | **107.35 ± 3.941** | **14.25 ± 0.920** |

Values are the means of triplicate experiments. Results are reported as mean ± SEM, VWR = varieties of tef with N rates (kg N ha$^{-1}$), **GMK, GMB, KMA, G.MV, G.MR** = grand means of **Kora (K), Boset (B) and Asgori (A)** tef varieties, and N rates respectively. And, (K0, K30, K60, K90, K120), (B0, B30, B60, B90, B120), and (A0, A30, A60, A90, 120) combining with 0, 30, 60, 90 and 120 N rates (kg N ha$^{-1}$) for the treatment number of 1 to 15, and S. EM = Standards of errors of means. Means in the same column and within the same treatment category followed by different letter are significantly different as judged by LSD at p < 0.05. **Note: G.M of Kora, Boset and Asgori tef varieties are considered as means of tef varieties (over all N rates).**

[37] that Fe content of white tef varieties were in ranged of between 9.5–37.7 mg 100 g$^{-1}$. On the other hand, the Fe content of red tef grain ranged between 11.6 - > 150 mg 100 g$^{-1}$; which is not in line with the report of *Abebe* [39], that red tef variety had a higher Fe content than white tef varieties. The experimental soil contamination may be one factor for the increments of Iron content of tef varieties [39]. The mineral content of Iron is significantly affected by means of varieties (over all N rates), but not by N rates (over all means of tef varieties) (Table 4).

**Table 4. Mineral contents of tef grain as influenced by the interaction of tef varieties with different N fertilizers rates.**

| Treatments | Parameters | | |
|---|---|---|---|
| VWR | P (ppm) | Fe (mg 100 g$^{-1}$) | Ca (mg 100 g$^{-1}$) |
| K0 | 216.7$^{cde}$ ± 0.087 | 27.63$^{defg}$ ± 0.826 | 57.67$^{de}$ ± 0.729 |
| K30 | 216.9$^{bcde}$ ± 0.197 | 27.41$^{defg}$ ± 0.496 | 72.83$^{bc}$ ± 0.883 |
| K60 | 217.5$^{ab}$ ± 0.144 | 25.85$^{efg}$ ± 0.796 | 68.14$^{abcd}$ ± 0.942 |
| K90 | 216.9$^{bcde}$ ± 0.108 | 29.12$^{cde}$ ± 0.906 | 73.89$^{ab}$ ± 0.971 |
| K120 | 216.6$^{de}$ ± 0.232 | 28.02$^{def}$ ± 0.883 | 68.60$^{abcd}$ ± 0.189 |
| **G.MA** | **216.91 ± 0.229** | **27.606 ± 0.881** | **68.226 ± 0.192** |
| B0 | 216.6$^{e}$ ± 0.275 | 28.49$^{cd}$ ± 0.463 | 74.85$^{a}$ ± 0.978 |
| B30 | 217.4$^{ab}$ ± 0.101 | 28.11$^{def}$ ± 0.623 | 61.67$^{de}$ ± 0.924 |
| B60 | 217.8$^{a}$ ± 0.475 | 32.57$^{bc}$ ± 0.704 | 63.38$^{bcde}$ ± 0.298 |
| B90 | 216.4$^{e}$ ± 0.117 | 34.58$^{ab}$ ± 0.719 | 65.13$^{abcde}$ ± 0.689 |
| B120 | 216.7$^{cde}$ ± 0.060 | 26.96$^{defg}$ ± 0.691 | 64.63$^{abcde}$ ± 0.0783 |
| **G.MA** | **216.92 ± 0.402** | **30.14 ± 0.807** | **65.93 ± 0.829** |
| A0 | 217.2$^{abc}$ ± 0.271 | 23.33$^{g}$ ± 0.161 | 62.85$^{cd}$e ± 0.089 |
| A30 | 217.1$^{bcd}$ ± 0.202 | 30.25$^{cd}$ ± 0.708 | 62.46$^{cde}$ ± 0.109 |
| A60 | 217.2$^{abcd}$ ± 0.43 | 24.22$^{f}$ ± 0.184 | 57.09$^{e}$ ± 0.029 |
| A90 | 216.7$^{cde}$ ± 0.266 | 36.49$^{ab}$ ± 0.807 | 58.84de ± 0.106 |
| A120 | 217.3$^{abc}$ ± 0197 | 37.39$^{a}$ ± 0.781 | 58.49de ± 0.149 |
| **G.MA** | **217.10 ± 0.270** | **30.334 ± 0.709** | **59.95 ± 0.245** |
| Means of varieties (over all N rates) | | | |
| Kora | 216.91$^{a}$ ± 0.101 | 22.60$^{b}$ ±0.638 | 68.22$^{a}$ ± 1.70 |
| Boset | 216.92$^{a}$ ± 0.101 | 30.14$^{a}$ ± 0.638 | 65.93$^{a}$ ± 1.70 |
| Asgori | 217.10$^{a}$ ± 0.101 | 30.38$^{a}$ ± 0.638 | 59.95$^{b}$ ± 1.70 |
| **G.MV** | **216.98 ± 0.507** | **27.71 ± 0.381** | **64.70 ± 1.87** |
| N rates (Over all means of tef varieties) | | | |
| 0 | 216.78$^{bc}$ ± 0.293 | 28.08$^{a}$ ± 0.128 | 58.07$^{a}$ ± 0.801 |
| 30 | 217.14$^{ab}$ ± 0.200 | 28.19$^{a}$ ± 0.208 | 60.84$^{a}$ ± 0.708 |
| 60 | 216.78$^{ab}$ ± 0.362 | 31.01$^{a}$ ± 0.807 | 59.07$^{a}$ ± 0.184 |
| 90 | 217.14$^{c}$ ± 0.204 | 30.83$^{a}$ ± 0.609 | 60.05$^{a}$ ± 0.419 |
| 120 | 216.78$^{bc}$ ± 0.244 | 31.08$^{a}$ ± 0.809 | 61.73$^{a}$ ± 0.147 |
| **G.MR** | **217.14 ± 0.307** | **29.838 ± 0.709** | **59.95 ± 0.909** |

Values are the means of triplicate experiments. Results are reported as mean ± SEM, VWR = varieties of tef with N rates (kg N ha$^{-1}$), **G.MK, G.MB, G.MA, G.MV, G. MR** = grand means of **Kora (K), Boset (B) and Asgori (A)** tef varieties, and N rates respectively. And, **(K0, K30, K60, K90, K120), (B0, B30, B60, B90, B120), and (A0, A30, A60, A90, 120)** combining with 0, 30, 60, 90 and 120 N rates (kg N ha$^{-1}$) for the treatment number of 1 to 15. And S. EM = Standards of errors of means, P = Phosphorus, Fe = Iron and Ca = Calcium and; means in the same column and within the same treatment category followed by different letter are significantly different as judged by LSD at p < 0.05. **Note: G.M of Kora, Boset and Asgori tef varieties are considered as means of tef varieties (over all N rates).**

## Calcium (Ca)

Calcium (Ca) content was significantly (P < 0.05) affected by tef varieties and the interaction effects (Table 4). Both Kora and Boset white tef varieties had a better calcium contents as compared to Red Asgori tef varieties (Table 4). The maximum (73.89, 74.85, and 62.85) mg 100 g$^{-1}$ Ca contents were obtained at plots with 90 (Kora), 0 (Boset and Asgori) kg N ha$^{-1}$, while the minimum (57.67, 61.67, and 57.09) mg 100 g$^{-1}$ were recorded at 0, 30 and 60 kg N ha$^{-1}$ of Kora, Boset and Asgori, respectively. The ranges of Ca content for both white tef varieties of Kora and Boset were found between 57. 67 and 74.85 mg 100 g$^{-1}$. On the other hand, the Ca content of Asgori was found between 57.09 and 62.85 mg 100 g$^{-1}$ which was the in line with

report of Baye [37]. With no consideration of nitrogen fertilizer rates, Kora variety had maximum (68.22) calcium content followed by Boset tef variety (65.93 mg 100 $g^{-1}$). Asgori tef variety has maximum Ca content (59.95 mg 100 $g^{-1}$) which is not in line with Abebe et al. [39] where red tef varieties have a better Ca contents.

## Phosphorus content of tef grain

The analysis of variance indicated that the P content of tef grain was significantly (P < 0.05) affected by N rates, but not by tef varieties. The present result tells us that, both white seeded tef varieties (Kora and Boset) and the brown seeded tef variety (Asgori) had almost the same P content. Abebe [39, 40] reported that red tef grain has higher P content which is not the same with the current study. Regardless of the tef varieties, **60** kg N $ha^{-1}$ has a maximum P content followed by 30 kg N $ha^{-1}$. This showed that nitrogen fertilizers affect P contents of tef grain. In addition to this, nitrogen fertilizer levels (overall average values of tef varieties) had significant effects on phosphorus content.

## Effects of nitrogen fertilizer rates and tef varieties on sensory quality of injera

The analysis of sensory qualities and acceptability of tef injera made from three tef varieties as influenced by different N fertilizer rates are presented in (Table 5). The sensory evaluation scored values for all sensory attributes were determined after 24 h that after the injera was baked using 15 semi-trained panelists.

## Sensory evaluation of injera

### Color

The analysis of variance indicated that, the color of injera was significantly (P < 0.05) affected by tef varieties and N fertilizer rates. The color of injera was slightly decreased with the increase of the first three N fertilizer rates. However, Kora had a maximum (8.27) color value of injera at the control plot (K0) and rated as like-very much. The grand means of a variety of Kora (G.MK) (6.63), Boset (G. MB) (6.53), and Asgori (G.MA) (6.04) showed that the color of baked injera was slightly affected by tef varieties. Average means of total nitrogen showed that the color of injera was maximum at the control plot (without nitrogen fertilizers) but it decreased at the first. On the other hand, the color of injera slightly increased at 120 kg $ha^{-1}$. Nitrogen fixation may negatively affect nitrogen availabilities for plant use. $CO_2$ gas bubbled leads to the formation of injera eye on its surface had honeycomb-like structure [41]. Injera having uniformly distributed eyes on its top surface of injera without blind spots tells us the injera is good and it affects by the consumers perception [42].

### Flavor

Interaction effect of tef varieties with N fertilizer rates significantly (P < 0.05) affected the flavor of injera. Gu et al. [13] reported that increasing N fertilizer application affects the flavor of a food product. Kora (7.1) and Boset (6.07) had a better color of injera as comparing with Asgori (5.73) and rated as neither like nor a dislike, like-slightly, and like moderately, respectively. Consumer's perception affects the sensorial qualities of injera which had significant product preferences. The average mean of Kora (G.MK) 6.13, Boset (G. MB) 6.05 and Asgori (G.MA) 6.04 had rated values of like-slightly, and means of N rate showed that application of all N rates except 90 kg N $ha^{-1}$ (neither like nor dislike) had the same effects on the flavor of injera.

**Table 5. Sensory acceptability test result of injera of tef varieties as influenced by five different N fertilizer rates.**

**Treatment Sensory attributes**

| VWR | Color | Flavor | Texture | Taste | FC | ES | ED | TBS | OACC |
|---|---|---|---|---|---|---|---|---|---|
| K0 | 8.27[a] ± 0.182 | 7.13[a] ± 0.336 | 6.87[a] ± 0.401 | 7.27[a] ± 0.358 | 7.00[a] ± 0.390 | 7.27[a] ± 0.419 | 6.73[ab] ± 0.511 | 6.87[a] ± 0.506 | 7.80[a] ± 0.355 |
| K30 | 7.07[abc] ± 0.300 | 5.93[ab] ± 0.442 | 5.93[ab] ± 0.408 | 6.13[ab] ± 0.559 | 6.07[a] ± 0.581 | 7.27[a] ± 0.419 | 7.33[a] ± 0.287 | 6.60[a] ± 0.486 | 7.47[a] ± 0.350 |
| K60 | 5.73[c] ± 0.530 | 6.00[ab] ± 0.447 | 6.13[ab] ± 0.363 | 6.80[ab] ± 0.223 | 6.27[a] ± 0.565 | 5.87[abc] ± 0.487 | 6.27[ab] ± 0.371 | 6.67[a] ± 0.333 | 6.87[a] ± 0.335 |
| K90 | 5.87[bc] ± 0.559 | 5.53[b] ± 0.487 | 5.93[ab] ± 0.442 | 5.87[c] ± 0.363 | 5.93[a] ± 0.371 | 6.33[abc] ± 0.532 | 6.47[ab] ± 0.363 | 6.87[a] ± 0.363 | 6.87[a] ± 0.456 |
| K120 | 6.20[bc] ± 0.460 | 6.07[ab] ± 0.483 | 6.07[ab] ± 0.521 | 6.67[ab] ± 0.398 | 6.33[a] ± 0.410 | 6.67[abc] ± 0.252 | 6.33[ab] ± 0.303 | 6.87[a] ± 0.256[a] | 7.20[a] ± 0.312 |
| **G.MK** | **6.63[a] ± 0.217** | **6.13[a] ± 0.202** | **6.19[a] ± 0.191** | **6.55[a] ± 0.191** | **6.32[a] ± 0.210** | **6.68[a] ± 0.198** | **6.63[a] ± 0.170** | **6.77[a] ± 0.175** | **7.24[a] ± 0.168** |
| B0 | 6.60[bc] ± 0.375 | 6.07[ab] ±0.383 | 6.20[ab] ±0.428 | 6.80[ab] ±0.500 | 6.80[a] ±0.327 | 6.60[abc] ± 0.335 | 5.87[b] ±0.435 | 7.07[a] ±0.330 | 6.87[abc] ± 0.236 |
| B30 | 6.33[bc] ± 0.454 | 6.20[ab] ± 0.428 | 6.80[ab] ± 0.312 | 6.20[ab] ±0.518 | 6.20[a] ±0.449 | 6.47[abc] ± 0.456 | 6.33[ab] ± 0.398 | 6.47[a]± 0.446 | 6.93[bc] ± 0.316 |
| B60 | 6.47[bc] ± 0.487 | 6.47[ab] ±0.401 | 5.87[ab] ± 0.456 | 6.47[ab] ±0.336 | 5.80[a] ±0.393 | 5.60[c] ± 0.349 | 6.00[ab] ± 0.458 | 6.73[a] ± 0.371 | 6.60[abc] ± 0.349 |
| B90 | 5.93[bc] ± 0.521 | 5.40[c] ±0.533 | 6.07[ab] ± 0.419 | 6.73[ab] ±0.300 | 6.27[a] ±0.396 | 5.87[abc] ± 0.477 | 5.67[b] ± 0.433 | 6.40[a] ± 0.445 | 6.80[abc] ± 0.327 |
| B120 | 7.33[ab] ± 0.485 | 6.13[ab] ± 0.524 | 6.47[ab] ± 0.413 | 6.87[ab] ±0.376 | 6.60[a] ±0.363 | 6.60[abc] ± 0.400 | 6.40[ab] ± 0.423 | 7.40[a] ± 0.349 | 7.07[abc] ± 0.521 |
| **G. MB** | **6.53[a] ± 0.210** | **6.05[a] ± 0.203** | **6.28[a] ± 0.188** | **6.61[a] ± 0.82** | **6.33[a] ± 0.173** | **6.23[a] ± 0.184** | **6.00[b] ± 0.190** | **6.81[a] ± 0.176** | **6.85[ab] ± 0.158** |
| A0 | 6.00[bc] ± 0.447 | 5.73[ab] ± 0.345 | 5.40[b] ± 0.400 | 6.00[ab] ± 0.276 | 6.20[a] ± 0.355 | 6.20[abc] ± 0.38 | 6.13[ab] ± 0.413 | 6.13[a] ± 0.400 | 6.40[bc] ± 0.445 |
| A30 | 5.53[c] ± 0.551 | 6.27[ab] ± 0.419 | 5.87[ab] ± 0.446 | 6.27[ab] ± 0.442 | 6.80[a] ± 0.355 | 7.13[ab] ± 0.350 | 6.73[ab] ± 0.300 | 6.93[a] ± 0.345 | 7.20[abc] ± 0.296 |
| A60 | 6.20[bc] ± 0.460 | 6.00[ab] ± 0.390 | 5.87[ab] ± 0.307 | 6.00[ab] ± 0.324 | 6.07[a] ± 0.452 | 6.27[abc] ± 0.384 | 5.87[b] ± 0.435 | 6.00[a] ± 0.516 | 6.60[abc] ± 0.375 |
| A90 | 6.13[bc] ±0.524 | 6.27[ab] ±0.358 | 5.80[ab] ±0.460 | 5.87[ab] ±0.350 | 6.60[a] ± 0.434 | 6.40[abc] ± 0.412 | 5.73[b] ± 0.530 | 6.07[a] ± 0.539 | 6.13[c] ± 0.542 |
| A120 | 6.33[bc] ± 0.607 | 5.93[ab] ± 0.581 | 5.80[ab] ± 0.449 | 6.07[ab] ± 0.473 | 6.40[a] ± 0.476 | 5.73[abc] ± 0.605 | 6.27[ab] ± 0.483 | 6.27[a] ± 0.463 | 6.60[abc] ± 0.375 |
| **G.MA** | **6.04[b] ± 0.229** | **6.04[a] ± 0.165** | **5.75[a] ± 0.182** | **6.04[b] ± 0.182** | **6.41[a] ± 0.184** | **6.35[a] ± 0.197** | **6.15[ab] ± 0.195** | **6.28[a] ± 0.203** | **6.59[b] ± 0.1854** |
| **N Rates (Overall tef varieties)** | | | | | | | | | |
| 0 | 6.96[a] ± 0.2461 | 6.31[a] ± 0.220 | 6.16[a] ± 0.248 | 6.69[a] ± 0.33 | 6.67[a] ± 0.208 | 6.89[ab] ± 0.224 | 6.24[ab] ± 0.262 | 6.69[a] ± 0.244 | 7.02[a] ± 0.256 |
| 30 | 6.31[ab] ± 0.268 | 6.13[a] ± 0.243 | 6.20[a] ± 0.230 | 6.20[a] ± 0.287 | 6.36[a] ± 0.270 | 6.95[a] ± 0.328 | 6.80[a] ± 0.198 | 6.67[a] ± 0.244 | 7.20[a] ± 0.171 |
| 60 | 6.13[ab] ± 0.270 | 6.16[a] ± 0.236 | 5.96[a] ± 0.215 | 6.42[a] ± 0.176 | 6.04[a] ± 0.270 | 5.91[c] ± 0.235[c] | 6.04[b] ± 0.240 | 6.47[a] ± 0.239 | 6.69[a] ± 0.217 |
| 90 | 5.98[b] ± 0.302 | 5.73[a] ± 0.268 | 5.93[a] ± 0.249 | 6.16[a] ± 0.201 | 6.27[a] ± 0.237 | 6.20[c] ± 0.271[bc] | 5.87[b] ± 0.257 | 6.44[a] ± 0.218 | 6.60[a] ± 0.258 |
| 120 | 6.62[ab] ± 0.304 | 6.04[a] ± 0.260 | 5.80[a] ± 0.259 | 6.53[a] ± 0.241 | 6.44[a] ± 0.237 | 6.33[ab] ± 0.258 | 6.33[ab] ± 0.231 | 6.84[a] ± 0.218 | 6.96[a] ± 0.222 |
| **G.M** | 6.40±0.127 | 6.08[a] ±0.110 | 5.75[a] ± 0.105 | 6.40[a] ± 0.103 | 6.36[a] ± 0.109 | 6.42[a] ± 0.112 | 6.26[a] ±0.108 | 6.62[a] ± 0.108 | 6.89[a] ± 0.099 |

**Values are the means of triplicate experiments.** Results were reported as **mean ± SEM**, **VWR** = varieties of tef with N rates (kg N ha$^{-1}$), **GMK, GMB, KMA** = grand means of Kora, Boset and Asgori tef varieties respectively, **FC** = folding capacity, **ES** = eye size, **ED** = eye distribution, **TBS** = top and bottom surface, **OACC** = over all acceptability of injera, **SEM** = standard error of mean and 0, **30, 60, 90 and 120** are N rates for the treatment number of 1 to 15. Means in the same column and within the same treatment category followed by different letter are significantly different as judged by LSD at $p < 0.05$. **Note: G.M of Kora, Boset and Asgori varieties are considered as means of varieties (over all N rates**

## Texture

Injera is Ethiopian traditional flat bread. It is soft and easily flexible to wrap and hold the sauce *(wot)* which makes it most preferable by consumers [42]. From the response of panelists and the analysis of variance, the texture of injera was not significant ($P < 0.05$) affected by tef varieties and N rates (Table 5). However, the interaction of both white seeded varieties of (Kora and Boset) and brown seeded Asgori variety had the same effects on the texture of injera and rated as like slightly, and neither like nor dislike, respectively. Gebrehiwot [43], reported that injera from only tef flour has no variation in texture values and this was not in line with the present study.

## Taste

The response of panelists showed that interaction of tef varieties with nitrogen fertilizer levels did not affected the taste values of injera. Injera from K90 and A90 had 5.87 taste score values are rated as neither like nor dislike. The taste values were ranged from 5.7 to 7.27 scores which rated as neither like nor dislike moderately. However, the current results were not in line with

[41] that taste value of red tef injera was rated as excellent. On the other hand, [43] reported that injera from *OE.Curvula* grain did not have significant effect. But the average means of tef varieties indicated that both white tef of Kora (6.55) and Boset (6.61) had the same taste scores. From the consumer's attitudes, injera from red tef had lower perception which leads to have fewer acceptances.

### Folding capacity (FC)

The folding capacity of injera was not significantly affected by varieties and N rates (Table 5). The maximum (7.00) and minimum (5.80) values of FC were obtained from KO and K90 and B60 which were rated as like-moderately and neither like nor dislike, respectively.

### Eye size (ES) and eye distribution (ED) of injera

Injera is large pancake-like bread prepared from cereal crops such as tef which is a traditional food in Ethiopia. Honeycomb-like holes produced on the surface of injera showed its basic quality characteristics. Both fermentation and carbon dioxide [44] bubbling during baking injera which is the main causes to produce eyes of injera on its top surfaces.

In the analysis of variance from the response of panelists, the eye size (ES) of baked injera was significantly ($P < 0.05$) affected by N fertilizer rates but not by varieties (Table 5). From this baked injera with different N fertilizer rates, except K60, B60, B90, and A120 (neither like nor dislike) and K0, K30, and A30 (like-moderately). On the other hand, the remaining baked injera was rated as like-slightly. This indicated that, the maximum grain yield (plants use the optimum N fertilizer) also has a significant connection with the changes in eye size. However, $CO_2$ have a positive effect on the formation of eye of injera. However it is not clearly stated that optimum nitrogen fertilizer have a negative important effect on carbon dioxide formation which contributes for the formations of eye size of injera.

On the other hand, eye distributions (ED) of injera were significantly ($P < 0.05$) affected by varieties and N rates. The ED of injera were rated neither like nor dislike and they were obtained at B0 (5.87), B90 (5.67), A60 (5.87), and A90 (5.73), but like moderately was recorded from K30 (7.33). The average mean of varieties showed that, ED of injera had significant differences between Kora (6.63), and Boset (6.00) varieties, while Asgori (6.15) had almost the same ED as Kora. The performance of ES and ED mainly depends on the formation and outflows of carbon dioxide gas [44]. In general, it is not clear to state that nitrogen fertilizer and escaping of carbon dioxide which is source of bubbles significantly affects the ES and ED of injera.

### Top and bottom surfaces (TBS) of injera

The appearances of the top and bottom surface of baked injera were not significantly ($p < 0.05$) affected by varieties and N rates (Table 5). From the response of panelists, all baked injera were rated as like-slightly except injera from B0 and B120 (like-moderately). The current result is in line with Yetneberk et al. [41] that injera made from tef flour had white top and bottom surfaces. On the other hand, nitrogen fertilizer rates did not improve or reduced witness of top and bottom surface of injera baked from tef grains.

### Overall acceptability of injera quality (OAIQ)

The overall acceptability of injera was significantly ($P < 0.05$) affected by varieties but not N rates. The maximum and minimum values of acceptability of injera were recorded at K0 (7.80) and A90 (6.13), respectively. From the response of panelists, almost all baked injera except K0, K30, K120, B120, and A30 (like-moderately) were rated as like-slightly (Table 5).

The average means of tef varieties showed that, the maximum OAIQ value were obtained from Kora (7.24) and Boset (6.85) tef varieties. Human's perception and appreciations to ward injera from red tef in the experimental study area makes difference between white and red tef varieties. On the other hand [43], reported that, overall acceptability of injera prepared from tef were accepted 100% and this is not agreed with the present study.

Considering all sensory attributes into account, there were a statistical difference among the injera prepared from the three varieties as influenced with nitrogen fertilizers and all treatments scored a mean rating above 6 (like-slightly) (Table 5) and this is indicative of the goodness of injera.

## Total number of holes and filtered eyes of injera

The color and number of holes of baked injera from three tef varieties interacts with different nitrogen fertilizer rates are determined using injera eye software (Fig 1) are presented in (Table 6).

**Table 6. Holes/ eyes and color values (CIE-l*ab) of injera of three tef varieties as influenced by N rates using *injera* eye software.**

| Treatment Parameters | | | | | |
|---|---|---|---|---|---|
| **Holes or eyes of injera** | | | **Color of injera** | | |
| VWR | Number of holes | Filtered eyes | Lightness L* | Redness a* | Yellowness b* |
| K0 | $14934.00^{ef} \pm 264.000$ | $7227.50^{fg} \pm 234.500$ | $55.20^{bc} \pm 0.173$ | $1.90^{c} \pm 0.036$ | $5.14^{e} \pm 0.621$ |
| K30 | $18350.00^{bcd} \pm 1119.000$ | $12476.50^{c} \pm 430.500$ | $54.64^{bcd} \pm 0.404$ | $0.84^{d} \pm 0.091$ | $3.64^{f} \pm 0.270$ |
| K60 | $16031.50^{de} \pm 586.500$ | $9919.00^{e} \pm 215.000$ | $56.82^{ab} \pm 0.197$ | $1.03^{d} \pm 0.086$ | $3.67^{f} \pm 0.872$ |
| K90 | $18300.00^{bcd} \pm 305.000$ | $12131.50^{c} \pm 84.500$ | $57.02^{ab} \pm 0.013$ | $1.21^{d} \pm 0.091$ | $3.72^{f} \pm 0.268$ |
| K120 | $18321.50^{bcd} \pm 982.500$ | $11823.50^{cd} \pm 416.500$ | $55.03^{bc} \pm 1.213$ | $0.76^{d} \pm 0.188$ | $4.51^{f} \pm 0.049^{e}$ |
| **G.MK** | $17187.40^{b} \pm 537.769$ | $10715.60^{b} \pm 659.992$ | $55.74^{a} \pm 0.488$ | $1.146^{b} \pm 0.140$ | $4.94^{b} \pm 0.729$ |
| B0 | $20028.50^{b} \pm 533.500$ | $8028.00^{f} \pm 423.000$ | $53.42^{de} \pm 0.140$ | $1.11^{d} \pm 0.288$ | $6.23^{d} \pm 0.200$ |
| B30 | $18930.50^{bc} \pm 1126.500$ | $10885.00^{de} \pm 57.000$ | $54.33^{cde} \pm 0.725$ | $1.06^{d} \pm 0.289$ | $4.93^{e} \pm 0.189$ |
| B60 | $27679.00^{a} \pm 1026.000$ | $15927.50^{a} \pm 199.500$ | $55.21^{bc} \pm 0.195$ | $0.75^{d} \pm 0.001$ | $3.90^{ef} \pm 0.003$ |
| B90 | $11172.00^{g} \pm 1622.000$ | $6348.50^{g} \pm 555.500$ | $56.62^{ab} \pm 0.151$ | $0.73^{d} \pm 0.186$ | $0.8^{5g} \pm 0.067$ |
| B120 | $20236.00^{b} \pm 166.000$ | $10656.00^{de} \pm 66.000$ | $53.97^{cde} \pm 0.032$ | $1.09^{d} \pm 0.108$ | $6.16^{d} \pm 0.342$ |
| **G.MB** | $19609.20^{a} \pm 1780.150$ | $10369.00^{b} \pm 1089.599$ | $54.71^{b} \pm 0.784$ | $0.949^{c} \pm 0.089$ | $4.41^{c} \pm 0.662$ |
| A0 | $20545.50^{b} \pm 266.500$ | $10968.00^{de} \pm 785.000$ | $47.59^{h} \pm 2.073$ | $11.15^{a} \pm 0.147$ | $7.37^{c} \pm 0.648$ |
| A30 | $12907.00^{fg} \pm 495.000$ | $7166.50^{fg} \pm 203.500$ | $53.4^{cde} \pm 0.183$ | $10.98^{ab} \pm 0.085$ | $9.95^{ab} \pm 0.266$ |
| A60 | $20519.00^{b} \pm 57.000$ | $13807.00^{b} \pm 252.000$ | $52.14^{efg} \pm 0.184$ | $10.63^{b} \pm 0.085$ | $9.84^{ab} \pm 0.266$ |
| A90 | $16934.00^{cde} \pm 87.000$ | $12767.00^{c} \pm 76.000$ | $52.51^{defg} \pm 0.000$ | $11.22^{a} \pm 0.000$ | $10.99^{a} \pm 0.000$ |
| A120 | $17917.00^{bcd} \pm 802.000$ | $11770.50^{cd} \pm 373.500$ | $50.66^{g} \pm 0.197$ | $11.05^{ab} \pm 0.012$ | $10.36^{a} \pm 0.010$ |
| **G.MA** | $17764.50^{b} \pm 979.849$ | $11295.80^{c} \pm 770.582$ | $51.26^{c} \pm 0.748$ | $11.01^{a} \pm 0.074$ | $9.704^{a} \pm 0.426$ |
| **N Rates (Overall means of tef varieties)** | | | | | |
| 0 | $18502.67^{b} \pm 1144.903$ | $8741.17^{d} \pm 757.572$ | $51.40^{d} \pm 1.489$ | $4.72^{a} \pm 2.041$ | $7.58^{a} \pm 0.587$ |
| 0 | $16729.17^{bc} \pm 1287.067$ | $10176.00^{c} \pm 1002.732$ | $54.13^{bc} \pm 0.320$ | $4.29^{b} \pm 2.116$ | $6.17^{b} \pm 1.223$ |
| 60 | $21409.83^{a} \pm 2166.755$ | $13217.83^{a} \pm 1117.176$ | $54.72^{b} \pm 0.874$ | $4.14^{b} \pm 2.056$ | $5.81^{ab} \pm 1.299$ |
| 90 | $15468.67^{c} \pm 1445.832$ | $10415.67^{c} \pm 1299.646$ | $55.45^{ab} \pm 1.250$ | $4.39^{b} \pm 2.163$ | $5.19^{ab} \pm 1.909$ |
| 120 | $18824.83^{b} \pm 560.057$ | $11416.67^{b} \pm 281.266$ | $53.22^{c} \pm 0.891^{c}$ | $4.30^{b} \pm 2.137$ | $7.01^{a} \pm 1.105$ |
| **G.MR** | **$18187.034 \pm 508.81$** | **$10793.468 \pm 295.087$** | **$53.784 \pm 0608$** | **$4.368 \pm 1.088$** | **$6.352 \pm 1.285$** |

**Values are means of replicate experiments.** Results were reported as **mean ± SEM**, **VWR** = varieties (tef) with N rates (kg N ha$^{-1}$), **G. MK, G. MB, K.MA** = grand means of Kora, Boset and Asgori respectively, **SEM** = standard error of mean, K = Kora, **B** = Boset and **A** = Asgori tef varieties and **0, 30, 60, 90 and 120** are N rates for the treatment number of **1 to 5 for each variety**. Means in the same column and within the same treatment category followed by different letters are significantly different as judged by LSD at P<0.05. **NOTE: G.M of Kora, Boset and Asgori are considered also as grand means of varieties (overall N rates).**

## Number of holes of injera

The total number of holes or eyes which present on the surface of injera were highly significant (P < 0.01) affected by N fertilizer rates and tef varieties (Table 6). On the other hand, both parameters did not showed increasing or decreasing trends. Obtaining a higher number of holes on the surface of injera may be due to the higher amount of carbon dioxide and gas bubbles in the fermented batter. There were difference between injera eye software and subjective panelists (human perceptions) on number of wholes of injera. Evaluations of sensory attributes using subjective methods can be varied from observer to observer. However, instrumental aspects such as injera eye software (Fig 1) become better as compared to subjective measurement. Numbers of wholes of injera were significantly affected by the average means of three tef varieties of Kora (G.MK), Boset (B.MB), and Asgori (A.MA).The maximum (19609.20) number of holes was recorded from Boset (G.MB) followed by Kora (G.MK) (17187.00) and Asgori (G.MA) (17764.50) varieties, respectively. Regardless of varieties, the average means N rate implies that the applications of 60 kg N ha$^{-1}$ produced the maximum (21409.83) number of holes of injera.

## Filtered eyes of injera

The filtered eyes of injera were highly significant (P < 0.01) affected by N fertilizer rates (Table 6). The average means of nitrogen fertilizer rates had almost the same effects on filtered eyes of on the surface of injera. For Kora tef variety, the minimum (7227.50) and maximum (12476.50) filtered eyes of injera were recorded from KO and K30, respectively. This indicated that the lower number of holes of injera was obtained from the control plot (without N fertilizer rates). On the other hand, for Bose tef variety the maximum (17927.50) and minimum (6348.50) filtered eyes of injera were obtained from B90 and B60, respectively. The ranges of filtered eyes of injera for Asgori tef variety were found between 7166.50 (A30) and 13807.00 (A60) respectively.

Grand means of varieties were significant on the filtered eyes of injera. Thus, Asgori variety had better-filtered eyes (11295.80) followed by both Kora (10715.60) and Boset (10369.00), respectively. This may show that carbon dioxide is produced during the fermentation process and plays a fundamental role in the formation of cellular structure leavened bread of Asgori [44]. Average mean of N rates indicated that, maximum (13217.83) filtered eyes on the surface of injera was obtained at 60 kg N ha$^{-1}$, while the minimum (8741.17) at 0 kg N ha$^{-1}$. Based on this, N fertilizer rate had positive significant effects on the productions of eyes on the surface of injera and the optimum N rate for this could be 60 g N ha$^{-1}$. The science of kinetics on batter fermentation and the nitrogen fertilizer and carbon dioxide as source of bubbles are not clear and they needs further investigation.

## Colors (CIE L *a*b*) values of injera

### Lightness (L) color of injera

CIE-l*a*b measures the lightness, redness, and yellowness color of injera in the respective order of L, a, and b. The lightness (L-value) determines the quality of injera. A good quality of injera had a better lightness values with lower redness and yellowness color scores.

The current result indicated that lightness color of injera were highly significant (P < 0.01) affected by N fertilizer rates and tef varieties (Table 6). [45], pointed out that lightness of injera the same ranges with the current results. Both Kora and Boset white tef varieties had the same lightness color values. In addition to this, the control plot and the plot with maximum (120 kg ha$^{-1}$) nitrogen fertilizer had the same lightness color values. This may be probably due to the

reason that as nitrogen fertilizer becomes increased and its availability decreased for plant uses. The average means of tef varieties indicated that the maximum and minimum lightness levels values injera were obtained from Kora (55.74) and Boset (54.71) followed by Asgori (51.26), respectively.

## Redness (a*) color of injera

The red color of injera was highly significantly (P < 0.01) affected by varieties and significantly (P < 0.05) affected by N rates and the interaction with varieties (Table 6). Both white seeded tef varieties had the same red color as result of the application of nitrogen fertilizer rates. This showed that, the color of injera depends on the grain color, which is the genetic variation of the varieties.

The maximum and minimum red colors of injera were recorded from the grand means of Asgori (11.01), Kora (1.46), and Boset (0.949), respectively. The higher the red color value, the lower the acceptance of injera. The genetic difference of tef varieties also a source of variation for the acceptance of injera. In addition to this, the red color of injera from Asgori tef variety may be due to the genetic difference. Despite of the genetic variations, the societies awareness also affects the acceptance of the product that injera from white tef varieties had a better acceptance as compared to injera from red tef variety.

## Yellowness (b*) color of injera

The yellowness color of injera was highly significantly (P < 0.01) affected by varieties and N rates including the interaction effects (Table 6).

Yellow color of injera were ranged from 0.89 (B90) to 11.99 (A90). Both white color tef varieties of Kora and Bose had maximum yellowness color at K0 (5.14) and B0 (6.32), respectively. Both the control plot and plot with maximum nitrogen fertilizer had maximum yellowness color of injera. This may be due to the reason that as level of nitrogen increased, its availabilities become decreased. In addition to this, it may be also due to the nitrogen fixation and leaching process which affects nitrogen availabilities. On the other hand, injera from the red tef variety (Asgori) had maximum yellowness color as compared to those white tef varieties. This yellowness color of injera indicated that injera prepared from Asgori tef variety had the least acceptance as compared to Kora and Boset varieties. The average means of tef varieties indicated that Asgori tef variety had the maximum (9.74) yellowness color of injera followed by Boset (4.41) and Kora (4.41) tef varieties (Table 6).

## Conclusions

The average means of nitrogen fertilizer rates showed that protein contents were increased with nitrogen fertilizer rates. However, carbohydrate contents were decreased with nitrogen fertilizer rates. Tannin contents were only significantly affected by tef varieties and this may due probably to genetic differences among tef varieties. From tef varieties, Kora at the control plot (without N fertilizers) had a better color, flavor, texture, and taste values of injera. The average mean values of tef varieties had no significant difference in color, flavor, texture, and taste values of injera. On the other hand, except injera from Asgori tef variety had a significant difference in color and taste values. This may be the reason that injera prepared from red tef variety have less acceptance by the society. Thus, the society had lower awareness, that injera prepared from white tef had better acceptance as compared with red tef.

Analysis of injera using injera eye software showed that, nitrogen fertilizer rates and tef varieties interaction effects did not show significant changes in the color of injera. In general, genetic difference was the main source of color difference of injera. In sum up, it could not be

concluded that only nitrogen fertilizer rates had significant effects on sensory quality and color of injera. Therefore, further investigation could be need based on the recommended blended mineral fertilizers associated with soil type and microbial interactions and fermentation kinetics of the dough.

## Acknowledgments

The authors are grateful for Ethiopian Institution of Agricultural Research and Addis Ababa University for the provided laboratory facilities. And, the authors appreciated for Debrezeit agricultural research and the tef national research program for their delivering three tef varieties used for this study.

## Author Contributions

**Conceptualization:** Hayelom Berhe Dagnaw.

**Data curation:** Hayelom Berhe Dagnaw.

**Formal analysis:** Hayelom Berhe Dagnaw.

**Supervision:** Ashagrie Zewdu Woldegiorgis, Kebebew Assefa Kebede.

**Validation:** Hayelom Berhe Dagnaw, Ashagrie Zewdu Woldegiorgis, Kebebew Assefa Kebede.

**Visualization:** Hayelom Berhe Dagnaw.

**Writing – original draft:** Hayelom Berhe Dagnaw.

**Writing – review & editing:** Kebebew Assefa Kebede.

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
