## [Decision Letter · Decision Letter 0]

20 Jun 2022

PONE-D-22-09743Influence of Nitrogen Fertilizer Rate and Variety on Tef [Eragrostis tef (Zucc.) Trotter] Nutritional Composition and Sensory Quality of a Staple bread (Injera)PLOS ONE

Dear Dr. Dagnaw,

Thank you for submitting your manuscript to PLOS ONE. After careful consideration, we feel that it has merit but does not fully meet PLOS ONE’s publication criteria as it currently stands. Therefore, we invite you to submit a revised version of the manuscript that addresses the points raised during the review process.

We look forward to receiving your revised manuscript.

Kind regards,

Shahbaz Khan, PhD

Academic Editor

PLOS ONE

Journal Requirements:

“The first author of this research article is Hayelom Berhe Dagnaw. I am working at Ethiopian Institute of Agricultural Research, Addis Ababa, Ethiopia. I graduated my M.Sc. degree in food Science and Nutrition from Addis Ababa University, Ethiopia. 

On the other hand, the second author (supervisor) of this research article was Ashagrie Zewdu Woldegiorgis (Ph.D.) with the position of associate professor. He works at Addis Ababa University, center for Food Siena and Nutrition.

The third author (supervisor) of this research article was Kebebew Asefa (Ph.D.), plant breeder, with the position of professor (senior researcher). He works at Ethiopian Institute of Agricultural Research, Addis Ababa, Ethiopia. The funders of this research article were Ethiopian Institute of Agricultural Research (EIAR) and Addis Ababa University (AAU). The URL of each funder website (EIAR) and AAU are, www.eiar.gov.et, and www.aau.edu.et, respectively. The funders had no role in study design, data collection and analysis, decision to publish, or preparation of the manuscript.”

 “We would like to give great gratitude and appreciation to the Ethiopian Institute of Agricultural 560 Research (EIAR) and Addis Ababa University (AAU) for supported this research financially.”

6. We note that you have referenced (ie. Bewick et al. [5]) which has currently not yet been accepted for publication. Please remove this from your References and amend this to state in the body of your manuscript: (ie “Bewick et al. [Unpublished]”) as detailed online in our guide for authors

Additional Editor Comments:

Dear Authors,

You are requested to submit the revised version of manuscript improved according the comments and suggestions raised by the reviewers' team.

Thanks.

Reviewers' comments:

Reviewer's Responses to Questions

**Comments to the Author**

1. Is the manuscript technically sound, and do the data support the conclusions?

Reviewer #1: Yes

Reviewer #2: Yes

Reviewer #3: Partly

2. Has the statistical analysis been performed appropriately and rigorously? 

Reviewer #1: Yes

Reviewer #2: Yes

Reviewer #3: No

3. Have the authors made all data underlying the findings in their manuscript fully available?

Reviewer #1: Yes

Reviewer #2: Yes

Reviewer #3: Yes

4. Is the manuscript presented in an intelligible fashion and written in standard English?

Reviewer #1: No

Reviewer #2: No

Reviewer #3: No

5. Review Comments to the Author

Reviewer #1: The survey reports an interesting topic that points out the influence of nitrogen fertilizer rate and variety on Tef [Eragrostis tef (Zucc.) Trotter] nutritional composition and sensory quality of a staple bread (injera). Despite that the study didn’t detected significant improvements with increasing N rates; however, scientifically, reporting these findings is very important and crucial. The manuscript presents a huge gap in its standard English; accordingly, I suggest passing it to a native English speaker for an extended editing. The Financial Disclosure and Ethics Statement should be adjusted according to the journal’s guidelines.

The Abstract part should be re-written due to an extensive English language issue. The majority of the keywords fit. The Introduction part is badly written in standard English; accordingly, it should be reformulated in a better language style. The Materials and methods part shows numerous needed adjustments in terms of sentences standard English formulation. Moreover, most methods especially related to laboratory analysis lack explanation, therefore they should be added. The Results and discussion part needs major adjustments mainly related to the scientific analysis of obtained findings. However, the statistical approach adopted is correctly performed. On the other hand, I suggest a major revision or re-writing this whole part as it shows uncountable linguistic mistakes and the need for a wide range of sentences reformulation. Regarding the Conclusions part, it summarizes the study’s findings and suggests further research. Only minor adjustments are required related to sentences reformulation.

Based on the above and below detailed explanation, recommendations and suggestions, I think that the manuscript shows a certain merit to be published in “PLoS One” once all suggestions and recommendations are fully taken into consideration and well addressed. Detailed comments, suggestions and recommendations are found within the attached file.

Reviewer #2: Abstract

Line 15, 18: Avoid using acronyms in summary.

Keywords

Line 31: avoid repeating the same ones in the title, (CIEL*a*b) Space of Color? could you specify something more general in relation to the CIEL*a*b, it is not clear what CIEL?

Introduction

Line 38-39: It is not necessary to cite so many references at this point, just keep the most current one.

Line 49: productivity of tef is 1.664 t/ha

Line 50 to 54: Improve the wording and focus on the importance of nitrogen in cereals.

Line 68-69: it is narrated as a discussion, not as an introduction to the work.

-There is a need to improve the connections between paragraphs and to write as an introduction to the topic, unclear and poorly structured.

Material and Methods

Line 119: @ typing mistake?

Line 128: 45 (Grades) typing mistake

Line 143: first time in extenso acronyms

Result and discussion

Objective: “this study mainly focused on the Influence of Nitrogen Fertilizer Rate and Variety Tef Nutritional Composition and Sensory Quality of a Staple bread (Injera)”

Line 160 to 172: does not respond to the objectives as described does not correspond to the results.

Line 177 to 181: Table 1, could be included as supplementary information in materials and methods.

- Focus on the most relevant results and responding to the objective set. If they were not significant, the variables that were not significant for the treatments used and the varieties evaluated can be listed in a single paragraph.

Conclusion

- Not responding to the objective concretely, it is a summary of the results obtained, not a conclusion based on the results.

Reviewer #3: The manuscript “Influence of Nitrogen Fertilizer Rate and Variety on Tef [Eragrostis tef (Zucc.) Trotter] Nutritional Composition and Sensory Quality of a Staple bread (Injera)” submitted by Dagnaw et al. in PONE described a large dataset and a good work conducted by team. However, this manuscript contains some flaws. Before reaching further decision the address and revision in manuscript is necessary:

1. Line no 13: imbalanced instead of unbalanced

2. Line no 15: the use of nitrogenous fertilizers

3. Line 19: ‘Crop attributes’ instead of ‘Recommended parameters’

4. Line 38-39: the number of reference need to reduce

5. Line 49: Mg ha-1 instead of t ha-1

6. Line 57: imbalanced instead of unbalanced

7. Line 68: no need of italic

8. Line 69: full form

9. Line 82-83: lowercase

10. Line 94: Olsen's method

11. Line 95: soil: water ratio

12. Line 100: triple super phosphate; lower case

13. Line 100: multiple sign ‘×’ not X

14. Line 108: superscript

15. Line 119: use degree sign;

16. Line 119: iron and other use lowercase

17. Line 162-63: part of materials and methods

18. Line 182-83: Tef grain as influenced by nitrogen fertilizer rates and varieties

19. Line 209: 9.88a ± 0.033 ; follow for all tables as applicable

20. Table no. 3 and 4 can be converted to Figures as the whole manuscript showing similar type of Tables.

21. Include some multivariate analyses for concluding valid recommendation among the nutritional and sensory indicator of Tef.

6. PLOS authors have the option to publish the peer review history of their article (what does this mean?). If published, this will include your full peer review and any attached files.

Reviewer #1: No

Reviewer #2: **Yes: **Cinthia Carolina Cazal Martinez

Reviewer #3: **Yes: **DR NIRMALENDU BASAK

---

## [Author Response · Author response to Decision Letter 0]

20 Dec 2022

Dear reviewers ; 

Here with I attached the reviewed manuscript (PONE-D-22-09743); that I have accepted all the comments you raised, And,I have tried to take corrections based on the recommendation in the manuscript.

---

## [Decision Letter · Decision Letter 1]

18 Jan 2023

PONE-D-22-09743R1Influence of Nitrogen Fertilizer Rate and Variety on Tef [Eragrostis tef (Zucc.) Trotter] Nutritional Composition and Sensory Quality of a Staple bread (Injera)PLOS ONE

Dear Dr. Dagnaw,

Thank you for submitting your manuscript to PLOS ONE. After careful consideration, we feel that it has merit but does not fully meet PLOS ONE’s publication criteria as it currently stands. Therefore, we invite you to submit a revised version of the manuscript that addresses the points raised during the review process.

We look forward to receiving your revised manuscript.

Kind regards,

Umakanta Sarker

Academic Editor

PLOS ONE

Reviewers' comments:

Reviewer's Responses to Questions

**Comments to the Author**

1. If the authors have adequately addressed your comments raised in a previous round of review and you feel that this manuscript is now acceptable for publication, you may indicate that here to bypass the “Comments to the Author” section, enter your conflict of interest statement in the “Confidential to Editor” section, and submit your "Accept" recommendation.

Reviewer #1: (No Response)

Reviewer #2: All comments have been addressed

2. Is the manuscript technically sound, and do the data support the conclusions?

Reviewer #1: Partly

Reviewer #2: Yes

3. Has the statistical analysis been performed appropriately and rigorously? 

Reviewer #1: Yes

Reviewer #2: Yes

4. Have the authors made all data underlying the findings in their manuscript fully available?

Reviewer #1: Yes

Reviewer #2: Yes

5. Is the manuscript presented in an intelligible fashion and written in standard English?

Reviewer #1: No

Reviewer #2: Yes

6. Review Comments to the Author

Reviewer #1: Authors made significant improvements to their manuscript related to the scientific analysis of the findings. However, the manuscript still shows serious and huge flows in its writing; it is extremely badly written in standard English. Therefore, I ask the authors to urgently pass their manuscript to a native English speaker for major editing and revision; otherwise, the manuscript cannot meet the high standards of Plos One journal.

Based on the above and below detailed explanation and the overall evaluation of the manuscript, I think that the manuscript needs extensive revision but shows a merit to be published in “PLoS One” once all suggestions and recommendations are fully taken into consideration and well addressed.

Abstract

1) Line 14: Kindly adjust as follow: “are considered”.

2) Line 24: Kindly adjust as follow: “but they decreased”.

3) Lines 24–25: “Asgori… injera”: The sentence is badly written in standard English; accordingly, kindly reformulate it.

4) Line 27: Kindly remove “had lower injera quality”.

1. Introduction

1) Line 39: “Tef… Ethiopia”: The sentence is badly written in standard English; accordingly, kindly reformulate it.

2) Lines 39–40: “Tef… wheat”: Same recommendation as in the previous comment.

3) Lines 41–44: “Tef… [5,6]”: Same recommendation as in the previous two comments.

4) Lines 44–45: “Tef… [7–9]”: Same recommendation as in the previous comments.

5) Lines 46–47: “Off… fertility”: Same recommendation as in the previous comment.

6) Line 48: Kindly adjust as follow: “compared to”.

7) Lines 50–53: “However… damage”: These sentences are badly written in standard English; accordingly, kindly reformulate them.

8) Lines 66–67: “Study… proteins”: Same recommendation as in the previous comment.

9) Line 73: Kindly remove “However”.

10) Line 76: Kindly replace “However” by “On the other hand”.

2. Materials and methods

1) Soil sample characterization, line 90: Kindly remove “And” from the beginning of the sentence.

2) Soil sample characterization, line 95: Kindly remove “by” before and after “using” and remove “And” from the beginning of the sentence.

3) Soil sample characterization, line 98: Kindly remove “And” from the beginning of the sentence.

4) Soil sample characterization, line 99: Kindly adjust as follow: “were added” and “allowed to stand”.

5) Soil sample characterization, line 100: Kindly remove “for”.

6) Soil sample characterization, line 103: Kindly remove “And” from the beginning of the sentence.

7) Soil sample characterization, lines 88–104: Kindly mention the full specification of the used instrumentation.

8) Experimental design, line 111: Kindly replace “is” by “was”.

9) Grain flour preparation, line 118: Kindly remove “first”.

10) Injera preparations, line 123: Kindly adjust as follow: “were mixed”.

11) Injera preparations, lines 125–126: Kindly adjust as follow: “℃”.

12) Injera preparations, line 127: Kindly adjust as follow: “were poured”.

13) Image analysis, line 135: Kindly adjust as follow: “To obtain”.

14) Image analysis, line 136: Kindly adjust as follow: “45°C”.

15) Image analysis, lines 136–137: “Before… walls”: The sentence is badly written in standard English; accordingly, kindly reformulate it.

16) Image analysis, lines 140–143: “Colors… on it”: Same recommendation as in the previous comment.

17) Laboratory analysis, Proximate analysis, lines 154–155: “Moisture… [25]”: Same recommendation as in the previous two comments.

18) Laboratory analysis, Proximate analysis, lines 155–159: “A crucible… (W3)”: Same recommendation as in the previous comments.

19) Laboratory analysis, Proximate analysis, lines 161–162: “The ash… 923.03”: Same recommendation as in the previous comments.

20) Laboratory analysis, Proximate analysis, line 164: Kindly adjust as follow: “were weighed”.

21) Laboratory analysis, Proximate analysis, lines 164–166: “Finally… charred”: The sentence is badly written in standard English; accordingly, kindly reformulate it.

22) Laboratory analysis, Proximate analysis, lines 171–175: “Air… 1 hour”: Same recommendation as in the previous comment.

23) Laboratory analysis, Proximate analysis, lines 181–183: “On the other… solvent”: Same recommendation as in the previous two comments.

24) Laboratory analysis, Proximate analysis, line 185: Kindly adjust as follow: “103℃”.

25) Laboratory analysis, Determination of mineral content, lines 194–201: “Iron… prepared”: These sentences are badly written in standard English; accordingly, kindly reformulate them.

26) Laboratory analysis, Determination of mineral content, lines 202–203: Kindly adjust as follow: “were prepared”.

27) Laboratory analysis, Determination of mineral content, line 209: Kindly adjust as follow: “were measured”.

28) Laboratory analysis, Determination of mineral content, lines 211–214: “After… completed”: The sentence is badly written in standard English; accordingly, kindly reformulate it.

29) Laboratory analysis, Determination of mineral content, line 215: Kindly adjust as follow: “were added”.

30) Laboratory analysis, Determination of mineral content, lines 216–223: “The solution… water”: These sentences are badly written in standard English; accordingly, kindly reformulate them.

31) Laboratory analysis, Determination of mineral content, lines 223 and 225: Kindly adjust as follow: “were prepared”.

32) Laboratory analysis, Determination of phytic acid content, line 229: Kindly adjust as follow: “Determination”.

33) Laboratory analysis, Determination of phytic acid content, lines 230–231: “The amount… [27]”: The sentence is badly written in standard English; accordingly, kindly reformulate it.

34) Laboratory analysis, Determination of phytic acid content, line 233: Kindly adjust as follow: “were added”.

35) Laboratory analysis, Determination of phytic acid content, lines 236–240: “A series… tubes”: These sentences are badly written in standard English; accordingly, kindly reformulate them.

36) Laboratory analysis, Determination of phytic acid content, lines 242–243: “Using… (1)”: Same recommendation as in the previous comment.

37) Laboratory analysis, Determination of phytic acid content, line 244: Kindly adjust the formula presentation following the journal’s guidelines.

38) Laboratory analysis, Determination of tannin content, line 245: Kindly adjust as follow: “Determination”.

39) Laboratory analysis, Determination of tannin content, line 249: Kindly adjust as follow: “were added”.

40) Laboratory analysis, Determination of tannin content, lines 250–251: “The samples… temperature”: The sentence is badly written in standard English; accordingly, kindly reformulate it.

41) Laboratory analysis, Determination of tannin content, lines 254–256: “Working… methanol”: Same recommendation as in the previous comment.

42) Laboratory analysis, Determination of tannin content, lines 256–259: “Five… 500 nm”: Same recommendation as in the previous two comments.

43) Laboratory analysis, Determination of tannin content, line 262: Kindly adjust the formula presentation following the journal’s guidelines.

44) Laboratory analysis, Determination of tannin content, line 263: Kindly adjust as follow: “were determined”.

45) Statistical analysis, lines 267–269: “The data… used”: The sentence is badly written in standard English; accordingly, kindly reformulate it.

3. Results and discussion

1) Soil physicochemical properties of the experimental site, lines 274–280: “The result… minerals”: These sentences are badly written in standard English; accordingly, kindly reformulate them.

2) Tef Grain as Influenced by Nitrogen Fertilizer Rates and Varieties, Moisture content, line 293: Kindly remove “G.”.

3) Tef Grain as Influenced by Nitrogen Fertilizer Rates and Varieties, Moisture content, line 296: Kindly replace “have” by “had” and adjust as follow: “as compared to”.

4) Tef Grain as Influenced by Nitrogen Fertilizer Rates and Varieties, Moisture content, line 297: Kindly adjust as follow: “(table 2)”.

5) Tef Grain as Influenced by Nitrogen Fertilizer Rates and Varieties, Ash content of tef grain, lines 299–301: “The analysis… variety”: These sentences are badly written in standard English; accordingly, kindly reformulate them.

6) Tef Grain as Influenced by Nitrogen Fertilizer Rates and Varieties, Protein content of tef grain, line 305: Kindly adjust as follow: “(Table 1)”.

7) Tef Grain as Influenced by Nitrogen Fertilizer Rates and Varieties, Protein content of tef grain, line 307: Kindly adjust as follow: “from that reported” and remove “G.”.

8) Tef Grain as Influenced by Nitrogen Fertilizer Rates and Varieties, Protein content of tef grain, lines 307–309: “Being… expected”: The sentence is badly written in standard English; accordingly, kindly reformulate it.

9) Tef Grain as Influenced by Nitrogen Fertilizer Rates and Varieties, Protein content of tef grain, line 309: Kindly adjust as follow: “significant”.

10) Tef Grain as Influenced by Nitrogen Fertilizer Rates and Varieties, Crude fat content, line 319: Kindly adjust as follow: “(Table 2)”.

11) Tef Grain as Influenced by Nitrogen Fertilizer Rates and Varieties, Crude fat content, line 320: Kindly remove “G.”.

12) Tef Grain as Influenced by Nitrogen Fertilizer Rates and Varieties, Crude fat content, lines 320–323: “The fat… [36]”: These sentences are badly written in standard English; accordingly, kindly reformulate them.

13) Tef Grain as Influenced by Nitrogen Fertilizer Rates and Varieties, Total Carbohydrate and energy content, line 328: Kindly adjust as follow: “(Table 2)”.

14) Tef Grain as Influenced by Nitrogen Fertilizer Rates and Varieties, Total Carbohydrate and energy content, lines 329–330: “On the other… increased”: The sentence is badly written in standard English; accordingly, kindly reformulate it.

15) Effects of Nitrogen Fertilizer Rates and Varieties on the Anti-nutritional Factor of Tef, Phytic acid content, line 337: Kindly adjust as follow: “(Table 3)”.

16) Effects of Nitrogen Fertilizer Rates and Varieties on the Anti-nutritional Factor of Tef, Phytic acid content, line 339: Kindly adjust as follow: “was found”.

17) Effects of Nitrogen Fertilizer Rates and Varieties on the Anti-nutritional Factor of Tef, Phytic acid content, lines 340–342: “The reason… contents”: The sentence is badly written in standard English; accordingly, kindly reformulate it.

18) Effects of Nitrogen Fertilizer Rates and Varieties on the Anti-nutritional Factor of Tef, Tannin content of tef grain, line 358: Kindly adjust as follow: “to reports”.

19) Effects of Nitrogen Fertilizer Rates and Varieties on the Anti-nutritional Factor of Tef, Tannin content of tef grain, lines 359–360: “Tannin… table 3”: The sentence is badly written in standard English; accordingly, kindly reformulate it.

20) Effects of Nitrogen Fertilizer Rates on Mineral Contents of Tef Grain, Iron (Fe), line 366: Kindly adjust as follow: “results are”.

21) Effects of Nitrogen Fertilizer Rates on Mineral Contents of Tef Grain, Iron (Fe), line 367: Kindly adjust as follow: “were in the range of”.

22) Effects of Nitrogen Fertilizer Rates on Mineral Contents of Tef Grain, Iron (Fe), line 370: “High… [39]”: The sentence is badly written in standard English; accordingly, kindly reformulate it.

23) Effects of Nitrogen Fertilizer Rates on Mineral Contents of Tef Grain, Iron (Fe), line 372: Kindly adjust as follow: “(Table 4)”.

24) Effects of Nitrogen Fertilizer Rates on Mineral Contents of Tef Grain, Calcium (Ca), line 385: Kindly adjust as follow: “(Table 4)” and “as compared to”.

25) Effects of Nitrogen Fertilizer Rates on Mineral Contents of Tef Grain, Calcium (Ca), lines 386–389: “The maximum… respectively”: The sentence is badly written in standard English; accordingly, kindly reformulate it.

26) Effects of Nitrogen Fertilizer Rates on Mineral Contents of Tef Grain, Calcium (Ca), line 393: Kindly adjust as follow: “maximum Ca content”.

27) Effects of Nitrogen Fertilizer Rates on Mineral Contents of Tef Grain, Calcium (Ca), line 394: Kindly adjust as follow: “observed with”.

28) Effects of Nitrogen Fertilizer Rates on Mineral Contents of Tef Grain, Calcium (Ca), lines 393–396: “On the other… varieties”: The sentence is badly written in standard English; accordingly, kindly reformulate it.

29) Effects of Nitrogen Fertilizer Rates on Mineral Contents of Tef Grain, Phosphorus content of tef grain, lines 401–402: “But… content”: The sentence is badly written in standard English; accordingly, kindly reformulate it.

30) Effects of Nitrogen Fertilizer Rates on Mineral Contents of Tef Grain, Phosphorus content of tef grain, line 404: Kindly adjust as follow: “in tef”.

31) Effects of Nitrogen Fertilizer Rates on Mineral Contents of Tef Grain, Phosphorus content of tef grain, line 405: Kindly replace “have” by “had”.

32) Effects of Nitrogen Fertilizer Rates and Tef Varieties on Sensory Quality of Injera, line 409: Kindly adjust as follow: “24 hours after”.

33) Sensory evaluation of injera, Color, lines 418–420: “Observing… rates”: The sentence is badly written in standard English; accordingly, kindly reformulate it.

34) Sensory evaluation of injera, Color, lines 421–425: “However… [42]”: Same recommendation as in the previous comment.

35) Sensory evaluation of injera, Flavor, lines 427–429: “Only… product”: Same recommendation as in the previous two comments.

36) Sensory evaluation of injera, Flavor, line 429: Kindly replace “have” by “had”.

37) Sensory evaluation of injera, Flavor, lines 431–432: “However… perceptions”: The sentence is badly written in standard English; accordingly, kindly reformulate it.

38) Sensory evaluation of injera, Flavor, line 435: Kindly remove “and rated as like-slightly”.

39) Sensory evaluation of injera, Texture, lines 437–438: “Injera… [42]”: The sentence is badly written in standard English; accordingly, kindly reformulate it.

40) Sensory evaluation of injera, Taste, lines 445–447: Same recommendation as in the previous comment.

41) Sensory evaluation of injera, Taste, lines 448–455: “However… acceptance”: Same recommendation as in the previous two comments.

42) Sensory evaluation of injera, Folding capacity (FC), line 457: Kindly adjust as follow: “(Table 5)”.

43) Sensory evaluation of injera, Folding capacity (FC), line 459: Kindly replace “are” by “were”.

44) Sensory evaluation of injera, Eye size (ES) and eye distribution (ED) of injera, lines 463–467: “Injera… injera”: These sentences are badly written in standard English; accordingly, kindly reformulate them.

45) Sensory evaluation of injera, Eye size (ES) and eye distribution (ED) of injera, line 469: Kindly adjust as follow: “(Table 5)”.

46) Sensory evaluation of injera, Eye size (ES) and eye distribution (ED) of injera, lines 478–480: “The ED… (7.33)”: The sentence is badly written in standard English; accordingly, kindly reformulate it.

47) Sensory evaluation of injera, Eye size (ES) and eye distribution (ED) of injera, lines 483–485: “Therefore… injera”: Same recommendation as in the previous comment.

48) Sensory evaluation of injera, Top and bottom surfaces (TBS) of injera, line 487: Kindly adjust as follow: “(p > 0.05)”.

49) Sensory evaluation of injera, Top and bottom surfaces (TBS) of injera, lines 489–492: “Yetnberk… varieties”: These sentences are badly written in standard English; accordingly, kindly reformulate them.

50) Sensory evaluation of injera, Overall acceptability of injera quality (OAIQ), line Sensory evaluation of injera, Overall acceptability of injera quality (OAIQ), lines 504–511: “Average… injera”: Same recommendation as in the previous comment.

51) Sensory evaluation of injera, Total number of holes and filtered eyes of injera, lines 513–515: Same recommendation as in the previous two comments.

52) Sensory evaluation of injera, Total number of holes and filtered eyes of injera, Number of holes of injera, lines 517–518: “The total… Table 6”: Same recommendation as in the previous comments.

53) Sensory evaluation of injera, Total number of holes and filtered eyes of injera, Number of holes of injera, lines 520–525: “As comparing… affected”: Same recommendation as in the previous comments.

54) Sensory evaluation of injera, Total number of holes and filtered eyes of injera, Number of holes of injera, line 527: Kindly adjust as follow: “implies that the application”.

55) Sensory evaluation of injera, Total number of holes and filtered eyes of injera, Filtered eyes of injera, lines 532–533: “The filtered… injera”: These sentences are badly written in standard English; accordingly, kindly reformulate them.

56) Sensory evaluation of injera, Total number of holes and filtered eyes of injera, Filtered eyes of injera, line 539: Kindly replace “to” by “and”.

57) Sensory evaluation of injera, Total number of holes and filtered eyes of injera, Filtered eyes of injera, lines 547–549: “However… investigation”: The sentence is badly written in standard English; accordingly, kindly reformulate it.

58) Sensory evaluation of injera, Colors (CIE L *a*b*) values of injera, Lightness (L) color of injera, line 553: Kindly adjust as follow: “determined the quality of injera”.

59) Sensory evaluation of injera, Colors (CIE L *a*b*) values of injera, Lightness (L) color of injera, lines 553–563: “Injera… respectively”: These sentences are badly written in standard English; accordingly, kindly reformulate them.

60) Sensory evaluation of injera, Colors (CIE L *a*b*) values of injera, Redness (a*) color of injera, line 565: Kindly adjust as follow: ‘significantly”.

61) Sensory evaluation of injera, Colors (CIE L *a*b*) values of injera, Redness (a*) color of injera, line 566: Kindly adjust as follow: “(Table 6)”.

62) Sensory evaluation of injera, Colors (CIE L *a*b*) values of injera, Redness (a*) color of injera, lines 566–567: “Nitrogen… varieties”: The sentence is badly written in standard English; accordingly, kindly reformulate it.

63) Sensory evaluation of injera, Colors (CIE L *a*b*) values of injera, Redness (a*) color of injera, lines 570–576: “From grand… injera”: Same recommendation as in the previous comment.

64) Sensory evaluation of injera, Colors (CIE L *a*b*) values of injera, Yellowness (b*) color of injera, line 594: Kindly adjust as follow: “(Table 6)”.

65) Sensory evaluation of injera, Colors (CIE L *a*b*) values of injera, Yellowness (b*) color of injera, lines 594–601: “The ranges… rates”: These sentences are badly written in standard English; accordingly, kindly reformulate them.

66) Sensory evaluation of injera, Colors (CIE L *a*b*) values of injera, Yellowness (b*) color of injera, line 602: Kindly replace “have” by “had”.

67) Sensory evaluation of injera, Colors (CIE L *a*b*) values of injera, Yellowness (b*) color of injera, lines 602–604: Kindly adjust as follow: “as compared to”.

4. Conclusions

1) Lines 608–609: “From the current… rates”: Kindly avoid the first voice form of the sentence and adopt the impersonal form instead.

2) Line 610: Kindly remove “was” before “increased” and “decreased”.

3) Lines 614–616: “Regardless… values”: The sentence is badly written in standard English; accordingly, kindly reformulate it.

4) Lines 616–617: “This may… society”: Same recommendation as in the previous comment.

5) Line 618: Kindly adjust as follow: “as compared to”.

6) Lines 618–624: “Analysis… dough”: These sentences are badly written in standard English; accordingly, kindly reformulate them.

Reviewer #2: Line 45, 73, 126, 330: review typographical or grammatical errors. Verify grammatical and typing errors throughout the document.

7. PLOS authors have the option to publish the peer review history of their article (what does this mean?). If published, this will include your full peer review and any attached files.

Reviewer #1: No

Reviewer #2: **Yes: **Cinthia C. Cazal Martinez

---

## [Author Response · Author response to Decision Letter 1]

26 Feb 2023

Dear all reviewers, as I attached the response to reviewers here using the application track, I would like to say thank you for your valuable comments and here I attached all the responses for all comments within the body (manuscript with/out track changes) and in separate for using the application track.

---

## [Decision Letter · Decision Letter 2]

3 Mar 2023

PONE-D-22-09743R2Influence of Nitrogen Fertilizer Rate and Variety on Tef [Eragrostis tef (Zucc.) Trotter] Nutritional Composition and Sensory Quality of a Staple bread (Injera)PLOS ONE

Dear Dr. Dagnaw,

Thank you for submitting your manuscript to PLOS ONE. After careful consideration, we feel that it has merit but does not fully meet PLOS ONE’s publication criteria as it currently stands. Therefore, we invite you to submit a revised version of the manuscript that addresses the points raised during the review process.

We look forward to receiving your revised manuscript.

Kind regards,

Umakanta Sarker

Academic Editor

PLOS ONE

Journal Requirements:

Additional Editor Comments:

Solve the typographical mistakes and spelling in the MS.

Change "ml" to "mL"

Add space after the symbol "°C". In some place, zero (0) was written in the place of degree symbol.

Add space between "words", "number and unit".

Add space before and after the symbol "≤", "≥", "±", "=", etc.

Superscript "-1" after ha.

Delete space before and after the symbol "/"

Reviewers' comments:

Reviewer's Responses to Questions

**Comments to the Author**

1. If the authors have adequately addressed your comments raised in a previous round of review and you feel that this manuscript is now acceptable for publication, you may indicate that here to bypass the “Comments to the Author” section, enter your conflict of interest statement in the “Confidential to Editor” section, and submit your "Accept" recommendation.

Reviewer #1: (No Response)

2. Is the manuscript technically sound, and do the data support the conclusions?

Reviewer #1: Yes

3. Has the statistical analysis been performed appropriately and rigorously? 

Reviewer #1: Yes

4. Have the authors made all data underlying the findings in their manuscript fully available?

Reviewer #1: Yes

5. Is the manuscript presented in an intelligible fashion and written in standard English?

Reviewer #1: No

6. Review Comments to the Author

Reviewer #1: Authors made significant linguistic improvements to their manuscript and are well thanked for that. However, the manuscript still minor adjustments in its language.

Based on the above and below detailed explanation and the overall evaluation of the manuscript, I think that the manuscript needs minor revision and shows a higher merit to be published in “PLoS One” once all suggestions and recommendations are fully taken into consideration and well addressed.

1. Introduction

1) Lines 46–47: “Tef… [7–9]”: The sentence is badly written in standard English; accordingly, kindly reformulate it.

2) Lines 49–50: “Off… fertility”: Same recommendation as in the previous comment.

3) Line 54: Kindly adjust as follow: “A study”.

4) Line 56: Kindly adjust as follow: “vegetative” and “affect”.

2. Materials and methods

1) Image analysis, lines 146–147: “The room… removed”: The sentence is badly written in standard English; accordingly, kindly reformulate it.

2) Image analysis, line 153: Kindly adjust as follow: “which was connected”.

3) Laboratory analysis, Proximate analysis, line 168: Kindly adjust as follow: “Crucibles”.

4) Laboratory analysis, Proximate analysis, line 172: Kindly adjust as follow: “after cooling”.

5) Laboratory analysis, Proximate analysis, line 181: Kindly adjust as follow: “Afterwards”.

6) Laboratory analysis, Determination of phytic acid content, lines 253–254: “The amount… [27]”: The sentence is badly written in standard English; accordingly, kindly reformulate it.

7) Laboratory analysis, Determination of phytic acid content, line 269: Kindly adjust as follow: “figured out”.

8) Laboratory analysis, Determination of tannin content, line 286: Kindly replace “becomes” by “was”.

9) Statistical analysis, lines 296–299: “And the statistical… ANOVA”: The sentence is badly written in standard English; accordingly, kindly reformulate it.

3. Results and discussion

1) Soil physicochemical properties of the experimental site, lines 304–305: “The current… (1997)”: The sentence is badly written in standard English; accordingly, kindly reformulate it.

2) Soil physicochemical properties of the experimental site, line 306: Kindly adjust as follow: “showed that”.

3) Soil physicochemical properties of the experimental site, line 308: Kindly adjust as follow: “was also”.

4) Soil physicochemical properties of the experimental site, line 310: Kindly replace "is" by "was".

5) Tef Grain as Influenced by Nitrogen Fertilizer Rates and Varieties, Ash content of tef grain, lines 332–333: “Bultosa… variety”: The sentence is badly written in standard English; accordingly, kindly reformulate it.

6) Tef Grain as Influenced by Nitrogen Fertilizer Rates and Varieties, Protein content of tef grain, lines 340–342: “However… one”: Same recommendation as in the previous comment.

7) Tef Grain as Influenced by Nitrogen Fertilizer Rates and Varieties, Crude fat content, lines 355–357: “The endosperm… [36]”: Same recommendation as in the previous two comments.

8) Tef Grain as Influenced by Nitrogen Fertilizer Rates and Varieties, Total Carbohydrate and energy content, lines 363–367: “Nitrogen… content”: Same recommendation as in the previous comments.

9) Effects of Nitrogen Fertilizer Rates and Varieties on the Anti-nutritional Factor of Tef, Tannin content of tef grain, line 395: Kindly adjust as follow: “to reports”.

10) Effects of Nitrogen Fertilizer Rates on Mineral Contents of Tef Grain, Iron (Fe), line 405: Kindly adjust as follow: “range”.

11) Effects of Nitrogen Fertilizer Rates on Mineral Contents of Tef Grain, Iron (Fe), line 411: Kindly adjust as follow: “Table”.

12) Effects of Nitrogen Fertilizer Rates on Mineral Contents of Tef Grain, Calcium (Ca), lines 433–434: Kindly replace “has” and “have” by “had” and adjust as follow: “content”.

13) Effects of Nitrogen Fertilizer Rates on Mineral Contents of Tef Grain, Phosphorus content of tef grain, line 444: Kindly adjust as follow: “in tef”.

14) Sensory evaluation of injera, Color, line 460: Kindly adjust as follow: “showed that”.

15) Sensory evaluation of injera, Color, lines 463–465: “Nitrogen… [41]”: These sentences are badly written in standard English; accordingly, kindly reformulate them.

16) Sensory evaluation of injera, Color, lines 467–469: “Injera… [42]”: Same recommendation as in the previous comment.

17) Sensory evaluation of injera, Flavor, lines 475–476: “However… preferences”: Same recommendation as in the previous two comments.

18) Sensory evaluation of injera, Taste, line 492: Kindly adjust as follow: “did not affect”.

19) Sensory evaluation of injera, Taste, lines 494–495: “Injera… dislike”: The sentence is badly written in standard English; accordingly, kindly reformulate it.

20) Sensory evaluation of injera, Taste, lines 497–499: “On the other… effect”: Same recommendation as in the previous comment.

21) Sensory evaluation of injera, Taste, line 501: Kindly adjust as follow: “acceptance”.

22) Sensory evaluation of injera, Eye size (ES) and eye distribution (ED) of injera, line 515: Kindly adjust as follow: “show”.

23) Sensory evaluation of injera, Eye size (ES) and eye distribution (ED) of injera, lines 516–517: “Both… surface”: The sentence is badly written in standard English; accordingly, kindly reformulate it.

24) Sensory evaluation of injera, Eye size (ES) and eye distribution (ED) of injera, lines 538–539: “In general… bubbles”: Same recommendation as in the previous comment.

25) Sensory evaluation of injera, Top and bottom surfaces (TBS) of injera, line 546: Kindly adjust as follow: “who reported”.

26) Sensory evaluation of injera, Top and bottom surfaces (TBS) of injera, line 548: Kindly adjust as follow: “reduce”.

27) Sensory evaluation of injera, Total number of holes and filtered eyes of injera, lines 574–576: These sentences are badly written in standard English; accordingly, kindly reformulate them.

28) Sensory evaluation of injera, Total number of holes and filtered eyes of injera, Number of holes of injera, lines 578–580: “The total… Table 6”: Same recommendation as in the previous comment.

29) Sensory evaluation of injera, Total number of holes and filtered eyes of injera, Number of holes of injera, lines 582–587: Same recommendation as in the previous two comments.

30) Sensory evaluation of injera, Total number of holes and filtered eyes of injera, Number of holes of injera, lines 588–590: “Numbers… (GMA)”: Same recommendation as in the previous comments.

31) Sensory evaluation of injera, Total number of holes and filtered eyes of injera, Filtered eyes of injera, line 598: Kindly adjust as follow: “(Table 6)”.

32) Sensory evaluation of injera, Total number of holes and filtered eyes of injera, Filtered eyes of injera, lines 599–600: “The average… injera”: The sentence is badly written in standard English; accordingly, kindly reformulate it.

33) Sensory evaluation of injera, Total number of holes and filtered eyes of injera, Filtered eyes of injera, line 617: Kindly adjust as follow: “and need”.

34) Sensory evaluation of injera, Colors (CIE L *a*b*) values of injera, Lightness (L) color of injera, lines 630–631: “This may… uses”: The sentence is badly written in standard English; accordingly, kindly reformulate it.

35) Sensory evaluation of injera, Colors (CIE L *a*b*) values of injera, Yellowness (b*) color of injera, line 667: Kindly remove “were”.

4. Conclusions

1) Line 689: Kindly remove “except”.

7. PLOS authors have the option to publish the peer review history of their article (what does this mean?). If published, this will include your full peer review and any attached files.

Reviewer #1: No

---

## [Author Response · Author response to Decision Letter 2]

13 Apr 2023

No comments, bu I have accepted all the comments and I took correction as I have been describe din the main body.

---

## [Decision Letter · Decision Letter 3]

27 Apr 2023

PONE-D-22-09743R3Influence of Nitrogen Fertilizer Rate and Variety on Tef [Eragrostis tef (Zucc.) Trotter] Nutritional Composition and Sensory Quality of a Staple bread (Injera)PLOS ONE

Dear Dr. Dagnaw,

Thank you for submitting your manuscript to PLOS ONE. After careful consideration, we feel that it has merit but does not fully meet PLOS ONE’s publication criteria as it currently stands. Therefore, we invite you to submit a revised version of the manuscript that addresses the points raised during the review process.

We look forward to receiving your revised manuscript.

Kind regards,

Umakanta Sarker

Academic Editor

PLOS ONE

Journal Requirements:

Reviewers' comments:

Reviewer's Responses to Questions

**Comments to the Author**

1. If the authors have adequately addressed your comments raised in a previous round of review and you feel that this manuscript is now acceptable for publication, you may indicate that here to bypass the “Comments to the Author” section, enter your conflict of interest statement in the “Confidential to Editor” section, and submit your "Accept" recommendation.

Reviewer #1: (No Response)

2. Is the manuscript technically sound, and do the data support the conclusions?

Reviewer #1: Yes

3. Has the statistical analysis been performed appropriately and rigorously? 

Reviewer #1: Yes

4. Have the authors made all data underlying the findings in their manuscript fully available?

Reviewer #1: Yes

5. Is the manuscript presented in an intelligible fashion and written in standard English?

Reviewer #1: No

6. Review Comments to the Author

Reviewer #1: Although authors made some linguistic improvements, the manuscript still needs to be linguistically revised by a native English speaker as numerous sentences are still rewritten in a bad manner.

Based on that, I think that the manuscript needs revision but shows a high scientific merit to be published in “PLoS One” once being adequately revised.

7. PLOS authors have the option to publish the peer review history of their article (what does this mean?). If published, this will include your full peer review and any attached files.

Reviewer #1: No

---

## [Editor Report · Decision Letter 4]

28 Sep 2023

PONE-D-22-09743R4Influence of Nitrogen Fertilizer Rate and Variety on Tef [Eragrostis tef (Zucc.) Trotter] Nutritional Composition and Sensory Quality of a Staple bread (Injera)PLOS ONE

Dear Dr. Dagnaw,

Thank you for submitting your manuscript to PLOS ONE. After careful consideration, we feel that it has merit but does not fully meet PLOS ONE’s publication criteria as it currently stands. Therefore, we invite you to submit a revised version of the manuscript that addresses the points raised during the review process.

ACADEMIC EDITOR:

The authors did not address the grammatical errors and typos. They must check the manuscript with a professional editing service. Furthermore, the author didn’t add any response letter. Supplementary documents have not been added to the manuscript. Revision must be with track changed mode for easy traceability of the corrections.

-There are many (hundreds) missing spaces between words, unit and unit, number and unit, number and °C, before and after the symbol “±”, “+”, “=”, “<”, “&”,  etc. throughout the whole MS including Table values, captions, and footnotes. Check carefully the whole MS to address these errors. 

Line 96: Change “< 2-mm” to “< 2 mm”. Change “ml” to “mL”. Follow this style throughout the whole MS including Table values, captions, and footnotes.

Line 122: Change “minutes” to “min”. Follow this style throughout the whole MS including Table values, captions, and footnotes.

Line 123: Change letter zero “48 hours for 28-30oC” to the symbol of degree “48 h for 28-30 °C”. Follow this style throughout the whole MS including Table values, captions, and footnotes.

Line 124: Change “2 - 3 hrs” to “2 - 3 h”. Follow this style throughout the whole MS including Table values, captions, and footnotes.

Line 224: Change “300ppm” to “300 ppm”. Follow this style throughout the whole MS including Table values, captions, and footnotes.

Line 231: Change “μg / g” to “μg/g”. Follow this style throughout the whole MS including Table values, captions, and footnotes.

Line 231: Change “Slpoe ×W × 3” to “Slope × W × 3”. Follow this style throughout the whole MS including Table values, captions, and footnotes.

Line 318: Change “P <0.001” to “P < 0.001”.

Line 320: Change “(table 3)” to “(Table 3)”. Many errors, address these errors.

Line 323: Change “mg 100 g-1” to “mg 100 g^-1^”. Many errors, address these errors.

Line 359: Change “able 4” to “Table 4”.

We look forward to receiving your revised manuscript.

Kind regards,

Umakanta Sarker

Academic Editor

PLOS ONE

Journal Requirements:

Additional Editor Comments:

The authors did not address the grammatical errors and typos. They must check the manuscript with a professional editing service. Furthermore, the author didn’t add any response letter. Supplementary documents have not been added to the manuscript. Revision must be with track changed mode for easy traceability of the corrections.

-There are many (hundreds) missing spaces between words, unit and unit, number and unit, number and °C, before and after the symbol “±”, “+”, “=”, “<”, “&”, etc. throughout the whole MS including Table values, captions, and footnotes. Check carefully the whole MS to address these errors.

Line 96: Change “< 2-mm” to “< 2 mm”. Change “ml” to “mL”. Follow this style throughout the whole MS including Table values, captions, and footnotes.

Line 122: Change “minutes” to “min”. Follow this style throughout the whole MS including Table values, captions, and footnotes.

Line 123: Change letter zero “48 hours for 28-30oC” to the symbol of degree “48 h for 28-30 °C”. Follow this style throughout the whole MS including Table values, captions, and footnotes.

Line 124: Change “2 - 3 hrs” to “2 - 3 h”. Follow this style throughout the whole MS including Table values, captions, and footnotes.

Line 224: Change “300ppm” to “300 ppm”. Follow this style throughout the whole MS including Table values, captions, and footnotes.

Line 231: Change “μg / g” to “μg/g”. Follow this style throughout the whole MS including Table values, captions, and footnotes.

Line 231: Change “Slpoe ×W × 3” to “Slope × W × 3”. Follow this style throughout the whole MS including Table values, captions, and footnotes.

Line 318: Change “P <0.001” to “P < 0.001”.

Line 320: Change “(table 3)” to “(Table 3)”. Many errors, address these errors.

Line 323: Change “mg 100 g-1” to “mg 100 g-1”. Many errors, address these errors.

Line 359: Change “able 4” to “Table 4”.

---

## [Author Response · Author response to Decision Letter 4]

12 Nov 2023

Line 96: Change “< 2-mm” to “< 2 mm”. Change “ml” to “mL”. Follow this style throughout the whole MS including Table values, captions, and footnotes.

Response: We thank the reviewer for his comment. In the current version of the article, I have changed all the recommenced comments in the whole manuscript. 

Line 122: Change “minutes” to “min”. Follow this style throughout the whole MS including Table values, captions, and footnotes.

Response: We thank the reviewer for his comment. In the current version of the article, that I changed all the recommended comments with their respect of the whole manuscript including table values captions and footnotes. 

Line 123: Change letter zero “48 hours for 28-30oC” to the symbol of degree “48 h for 28-30 °C”. Follow this style throughout the whole MS including Table values, captions, and footnotes.

Response: In the current version of the article, we discuss and modified all recommended comments in the current revised version of the manuscript. 

Line 124: Change “2 - 3 hrs” to “2 - 3 h”. Follow this style throughout the whole MS including Table values, captions, and footnotes.

Response: In the current of this revised version of the manuscript; I tried to respond and answer all the recommended comments throughout the whole manuscript. 

Line 224: Change “300ppm” to “300 ppm”. Follow this style throughout the whole MS including Table values, captions, and footnotes.

Response: In the current version of the revised manuscript I made all corrections regarding the given comments, the unit and their magnitude. 

Line 231: Change “μg / g” to “μg/g”. Follow this style throughout the whole MS including Table values, captions, and footnotes.

Response: For this comment regarding the unit in the article, I made corrections in the revised manuscript in all body of the manuscript regarding to this specific comments. 

Line 231: Change “Slpoe ×W × 3” to “Slope × W × 3”. Follow this style throughout the whole MS including Table values, captions, and footnotes.

Response: We thank the reviewer for the comment. In the current version of the article, I made the corrections regarding the equation in the revised manuscript throughout the whole manuscript including table values, captions and footnotes. 

Line 318: Change “P <0.001” to “P < 0.001”.

Response: We thank the reviewer for the comment. In the current version of the article, I changed the Stastical values of the “P” value to the standard format throughout the whole discuss the differences between targeting the different diseases.

Line 320: Change “(table 3)” to “(Table 3)”. Many errors, address these errors.

Response: We thank the reviewer for his comment. In the current version of the article, I changed the “Table format” throughout the whole body of the manuscript. 

Line 323: Change “mg 100 g-1” to “mg 100 g-1”. Many errors, address these errors.

Response: We thank the reviewer for his comment. In the current version of the article, I checked and made corrections regarding the superscript of the units throughout the whole body of the manuscript. 

Line 359: Change “able 4” to “Table 4”.

Response: We thank the reviewer for his comment. In the current version of the article, I made correction for the recommended comment regarding the standard Table format throughout the whole body of the manuscript.

---

## [Editor Report · Decision Letter 5]

23 Nov 2023

Influence of Nitrogen Fertilizer Rate and Variety on Tef [Eragrostis tef (Zucc.) Trotter] Nutritional Composition and Sensory Quality of a Staple bread (Injera)

PONE-D-22-09743R5

Dear Dr. Dagnaw,

We’re pleased to inform you that your manuscript has been judged scientifically suitable for publication and will be formally accepted for publication once it meets all outstanding technical requirements.

Kind regards,

Umakanta Sarker

Academic Editor

PLOS ONE
---

## [Editor Report · Acceptance letter]

19 Dec 2023

PONE-D-22-09743R5 

PLOS ONE

Dear Dr. Dagnaw, 

I'm pleased to inform you that your manuscript has been deemed suitable for publication in PLOS ONE. Congratulations! Your manuscript is now being handed over to our production team.

Kind regards, 

on behalf of

Professor Umakanta Sarker 

Academic Editor

PLOS ONE